behaviour, ecology, evolution

*bicyclus anynana*, lepidoptera, predation experiment, butterfly behaviour, wing overlap, mantid predators

**Authors for correspondence:**
Ian Z. W. Chan
e-mail: ianchan@nus.edu.sg
Antónia Monteiro
e-mail: antonia.monteiro@nus.edu.sg

†These authors contributed equally to this study.

# Predation favours *Bicyclus anynana* butterflies with fewer forewing eyespots

Ian Z. W. Chan[1,†], Zhe Ching Ngan[1,†], Lin Naing[1], Yueying Lee[1], V Gowri[1] and Antónia Monteiro[1,2]

[1]Department of Biological Sciences, National University of Singapore, 14 Science Drive 4 117557, Singapore
[2]Science Division, Yale-NUS College, 16 College Avenue West, 138527, Singapore

IZWC, 0000-0002-0973-7477; AM, 0000-0001-9696-459X

There are fewer eyespots on the forewings versus hindwings of nymphalids but the reasons for this uneven distribution remain unclear. One possibility is that, in many butterflies, the hindwing covers part of the ventral forewing at rest and there are fewer forewing sectors to display eyespots (covered eyespots are not continuously visible and are less likely to be under positive selection). A second explanation is that having fewer forewing eyespots confers a selective advantage against predators. We analysed wing overlap at rest in 275 nymphalid species with eyespots and found that many have exposed forewing sectors without eyespots: i.e. wing overlap does not constrain the forewing from having the same number or more eyespots than the hindwing. We performed two predation experiments with mantids to compare the relative fitness of and attack damage patterns on two forms of *Bicyclus anynana* butterflies, both with seven hindwing eyespots, but with two (in wild-type) or four (in Spotty) ventral forewing eyespots. Spotty experienced more intense predation on the forewings, were shorter-lived and laid fewer eggs. These results suggest that predation pressure limits forewing eyespot number in *B. anynana*. This may occur if attacks on forewing eyespots have more detrimental consequences for flight than attacks on hindwing eyespots.

## 1. Background

Colour patterns protect many organisms against predators [1]. Eyespots, 'roughly circular patterns with at least two concentric rings or with a single colour disc and a central pupil' [2], are a common protective pattern on many lepidopteran wings [3,4]: either intimidating predators (preventing them from attacking the prey) or deflecting predator attacks towards less important parts of the prey's body [5]. Eyespot effectiveness is dependent on multiple factors, e.g. eyespot size and number [6,7], behaviour such as rapid wing displays or stridulation [8], and predator type, as the same eyespot may appear intimidating to some predators but not to others [5].

Clusters of small eyespots on butterfly wing margins appear to make the area more conspicuous to predators and thereby deflect attacks onto themselves, away from vital body parts. In *Pararge aegeria*, which exhibits eyespot number polymorphism, males which spend more time displaying their wing patterns have more eyespots [9,10]. More directly, ventral eyespots in *Melanitis leda* and *Bicyclus anynana* deflect attacks from reptiles [11] and mantids [7], respectively. Furthermore, butterfly paper models with more conspicuous small ventral marginal eyespots are more readily attacked [6,12]. This increased conspicuousness, however, can confer fitness benefits because the eyespots draw attacks to less important body parts, as demonstrated in studies using live prey items [7].

Surveying museum specimens of 451 nymphalid species, Tokita *et al.* found twice as many eyespots on hindwings versus forewings on dorsal and ventral surfaces in males and females [13] (electronic supplementary material, figure S1).

*Proc. R. Soc. B* **288**: 20202840

(a) (b)

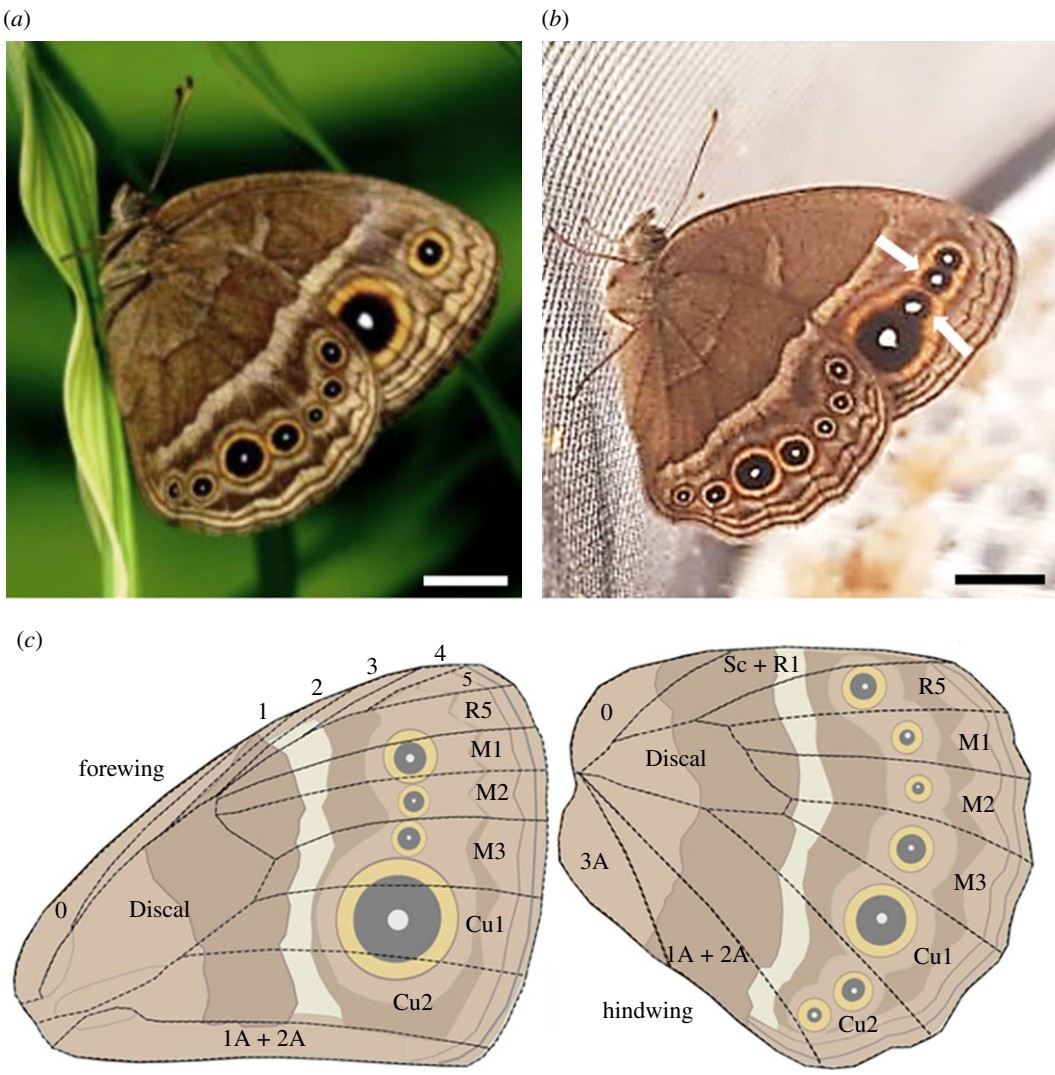

(c)

**Figure 1.** Examples of the Wt (*a*) and Spotty (*b*) *Bicyclus anynana* butterflies used in this study. Scale bars represent 4 mm. White arrows in (*b*) indicate the two additional forewing eyespots in the Spotty form. (*c*) In our microcosm experiment, damage from mantid attacks on butterfly wings were assigned to these labelled sectors of the fore and hindwings for later analysis. (Online version in colour.)

There are two potential explanations for this pattern, especially with respect to ventral eyespots which are important in anti-predator defence. First, when many butterflies perch, their hindwings cover part of the ventral forewing. Any eyespot on covered wing sectors would not be continuously visible to predators [14] and this wing overlap may therefore limit the number of forewing ventral eyespots that are maintained through positive selection (e.g. with a function in attack deflection). In other words, if a butterfly has fewer exposed forewing sectors than hindwing sectors carrying eyespots, it cannot have more exposed forewing eyespots than hindwing eyespots. Although covered eyespots may function in startle displays (e.g. [15,16]), this behaviour has not been documented in our study species *B. anynana*. Second, higher numbers of eyespots on the hindwings may be advantageous if they deflect predator attacks away from the forewings which are more important for flight. Here, we investigate these two hypotheses.

To study whether forewing ventral eyespot number is limited by wing overlap, we analysed images of live perching specimens of the species from Tokita *et al.* [13]. To investigate whether the pattern is maintained by predation pressure, we asked whether the attack behaviour of mantid predators, *Polyspilota* sp. and *Hierodula* sp., is influenced by variation in forewing eyespot number in a nymphalid butterfly,

*B. anynana*, which has eyespots on the margins of both forewings and hindwings. Each eyespot has a white centre [17] within a black disc and orange ring (figure 1*a*). The species has a variant, Spotty (figure 1*b*), found naturally at low frequencies, which differs from the wild-type (Wt) form in the presence of two extra eyespots on the forewing [18,19] that are visible at rest.

*Bicyclus* butterflies are likely to encounter *Polyspilota* mantids in the wild [20,21], while *Mycalesis* butterflies (a sister lineage with very similar wing patterns) are likely to encounter *Hierodula* mantids [22,23]. We hypothesize that mantids, common inhabitants of African [24] and Asian tropical forests [25], contributed to shaping the evolution of wing patterns in both taxa. We further hypothesize that, because the two additional forewing eyespots in Spotty *B. anynana* are likely to make this wing more conspicuous than the Wt forewing, more attacks would be directed at the forewing in Spotty [7], resulting in lower fitness. We tested this by exposing Spotty and Wt forms of *B. anynana* to mantid predators in two experiments: (i) a microcosm experiment to quantify the effects of mantid predation on the fitness of the two forms in terms of survival, fecundity and wing damage; and (ii) an arena-based predation experiment to observe which parts of the wings are targeted in mantid attacks.

## 2. Material and methods

### (a) Wing overlap analysis

We sourced images of the 451 species from Tokita *et al.* [13] in their natural resting positions from biodiversity databases, nature guides and wildlife photographers online. For each species, we noted which ventral forewing sectors are typically covered by the hindwing at rest. Because there can be variation within species and individuals, we surveyed as many images as possible per species and chose the most commonly observed degree of overlap. When there was a tie, we chose the more conservative observation, i.e. with more covered sectors. Hence, our estimate for the number of exposed forewing sectors at rest is likely to be conservative.

We then investigated whether exposed wing area limits forewing eyespot number to being lower than hindwing eyespot number through three analyses. First, in all species with eyespots, we divided the number of eyespots by the number of exposed sectors on each wing to investigate whether there are still fewer forewing versus hindwing eyespots after controlling for the number of exposed wing sectors. Second, we compared the number of exposed forewing versus hindwing eyespots in only those species which are not constrained to having fewer exposed forewing eyespots by wing overlap, i.e. these species have equal or more exposed forewing sectors (either with or without eyespots) than hindwing eyespots. Third, in all species with eyespots, we tested whether the absence of the M2 and M3 eyespots on the forewing (which differentiate the two *B. anynana* forms in our study), when homologues are present on the hindwing, can be accounted for by the frequency that these forewing sectors are covered. Assuming similar developmental constraints between the forewing and hindwing, any difference between the eyespot distributions would be a result of selection.

### (b) Experimental animals for predation experiments

For the microcosm experiment, male and female *Polyspilota* mantids were purchased from BugzUK (34 North Walsham Road, Norwich, UK) at the 6th instar and reared until adulthood for use. For the arena experiment, giant Asian mantis nymphs *Hierodula* sp. were trapped in a park along Upper Serangoon Road, Singapore. They were reared to adulthood in the laboratory and only female mantids were used. Their diet (fruit flies, grasshoppers and mealworms) was chosen to avoid exposure to butterflies or eyespots. Eggs of Wt and Spotty *B. anynana* were obtained from laboratory populations. Larvae were fed young maize *Zea mays* plants. The pupae of different sexes were separated to ensure virginity. Adults were fed pulped banana and water and used in experiments at between two to six days old.

### (c) Microcosm experiment set-up and analysis

We performed a microcosm experiment to investigate differences in survival, fecundity and wing damage in Wt and Spotty butterflies. Forty cylindrical net cages (50 cm diameter by 100 cm height), each containing 10 Wt or 10 Spotty *B. anynana* (five males and five females), were set up. For 20 cages (10 Wt and 10 Spotty), one mantid was introduced. Mantids were randomly assigned, while ensuring that each form was exposed to a similar number of male and female mantids. The other 20 cages served as controls. All cages were kept at 27°C and 60% relative humidity with a 12 L : 12 D cycle under a full-spectrum light source. A food source and an oviposition substrate were maintained until all butterflies were dead. Survival, fecundity and wing damage data were collected from all cages.

Survival and fecundity were measured by counting the number of live butterflies and eggs laid in each cage at the same time each day. Wing damage was quantified by examining the wings of dead butterflies for damage marks (holes, tears or missing chunks) and assigning scores to the damaged sectors (figure 1*c*). Each damage mark was worth a total of one point. When a mark occurred entirely within one sector, that sector alone was awarded the point. Where a mark was spread across multiple sectors, each sector shared the point equally (e.g. if a mark affected two wing sectors, each was assigned 0.5 points).

Data were analysed in R (v. 4.0.0) [26]. The survival data were analysed by a survival analysis (figure 2*a*) using the survival [27] and survminer [28] packages. Both constant and non-constant hazard functions were fitted and the model with the lowest Akaike information criterion value was chosen. Because mantid gender, which we expect to be important in experimental trials (those with mantids) [12], cannot be run in the same model as control trials (without a mantid), three separate models were run: (i) analysing all trials (control and experimental) with mantid presence/absence as the main effect to assess the effects of predation; (ii) analysing only control trials with butterfly form as the main effect to investigate the difference between forms in the absence of predation; and (iii) analysing only experimental trials with butterfly form and mantid gender as interacting main effects to assess differences between the two forms under predation. Where appropriate, cage number was included as a random effect to account for among-cage differences.

The fecundity data (figure 2*b*) were analysed using a Gaussian generalized linear mixed model (GLMM) via the glmmTMB package [29]. The response variable was the number of eggs laid per butterfly per day. As with the survival analysis, three separate models were run. Models were checked for adherence to error distribution assumptions using the DHARMa package [30].

Wing damage scores were plotted on two-way heat maps to compare damage on the two forms visually (figure 2*c*). The scores were also analysed using the mvabund package [31] to fit multivariate GLMMs where the response variable was the matrix of wing damage scores for each sector. This analysis detects overall differences in the number and distribution of damage marks between the forms. Models were fitted with a negative binomial error distribution, which was checked using the same package. Forewings and hindwings were analysed separately as they had different wing sectors. Similar to above, three sets of models were constructed, initially with form as the main explanatory variable and other potential sources of variation as secondary explanatory variables (e.g. cage number, mantid gender and average survival for each cage), and thereafter simplified stepwise by removing interactions or variables with a non-significant *p*-value. Where a variable had a *p*-value close to 0.05, we used the ANOVA function in base R [31] to compare how well the two models (i.e. before and after removing the variable in question) explained the data. If the amount of variance explained did not differ significantly, we removed the variable (as the simpler model could explain a similar amount of variance). Where the unsimplified model (i.e. with the variable) was significantly better, the variable was retained.

### (d) Arena experiment set-up and analysis

We performed an arena experiment to investigate differences in the wing areas targeted by mantids in Wt and Spotty butterflies. The wooden arena, covered with green poster board, was a square floor with a circular wall enclosing a rectangular ramp (46 cm length by 8 cm width) extending from a port in the wall to the centre of the arena, tilted upwards at an angle of 16° (similar to that used by Prudic *et al.* [7]; figure 3*a*). It was illuminated by two full-spectrum halogen tube lamps. Before trials, mantids and butterflies were starved for two days and one day, respectively. Immediately prior to trials, mantids were exposed to

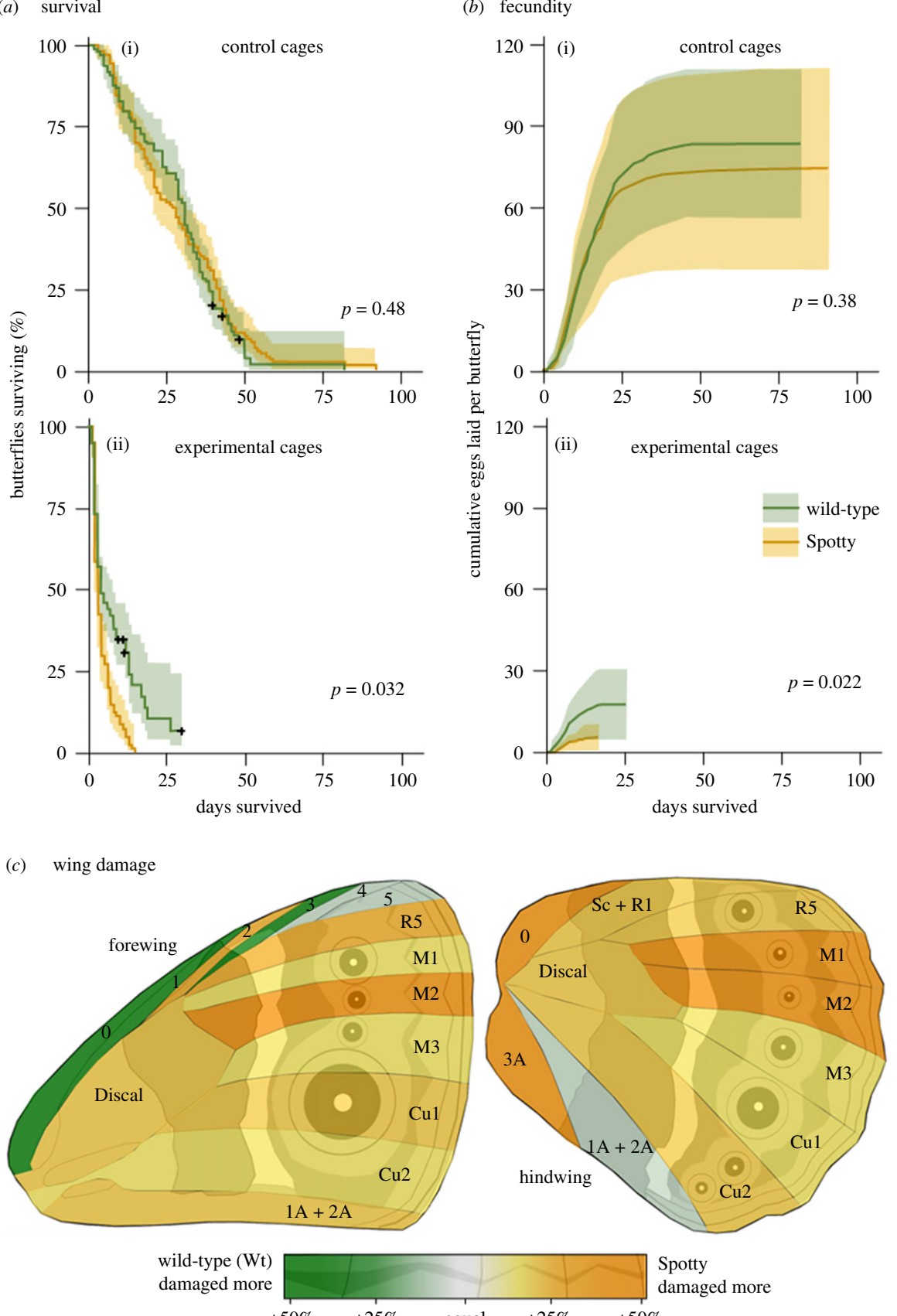

**Figure 2.** Results from the microcosm experiments. (*a*) The survival curves show that individuals in control cages (above) lived longer than those in experimental cages (below). Similar longevity between Wt and Spotty individuals in control cages, together with decreased longevity of Spotty versus Wt individuals in experimental cages, indicates that mantids preyed more intensely upon Spotty individuals. '+' indicates the censoring of a cage. (*b*) Similarly, Spotty fecundity was more significantly impaired by the mantid predators than Wt fecundity. Ranges in (*a*) and (*b*) represent 95% confidence intervals. (*c*) A two-way heat map with each wing sector coloured according to the form which suffered higher average damage scores in that sector (green for Wt and yellow for Spotty; only experimental trials shown). There are two clear patterns: first, the wings are predominantly yellow suggesting that Spotty individuals suffered more intense predation, and second the darker yellows tend to occur on the anterior sectors of both wings, suggesting that the mantids tended to direct more of their attacks towards the forward portions of both wings in Spotty.

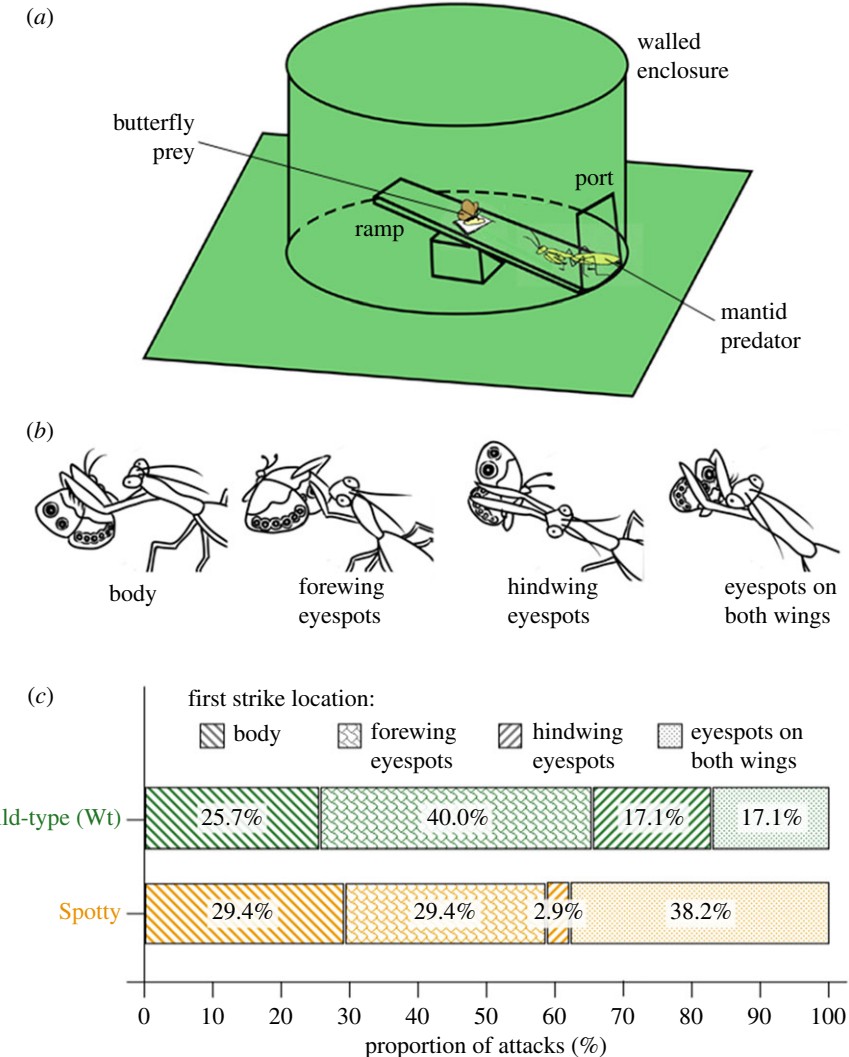

**Figure 3.** Results from the arena experiments. (*a*) Diagram of the arena used to ritualize the predator–prey interactions in our experiment. (*b*) The four categories of first strike locations, i.e. the part of the butterfly on which the mantid predator first struck during its attack. (*c*) Comparing Spotty to Wt, the results suggest that the mantids predominantly switched away from striking the hindwing eyespots alone in Wt (this category is much reduced in Spotty) to attacking the eyespots on both wings in Spotty (this category more than doubled to almost 40% in Spotty). The proportion of first strikes on the body in both forms remains similar. (Online version in colour.)

both *B. anynana* forms for 15 min by placing them into a small mesh cage between two other cages (one containing a Wt and the other a Spotty individual), during which the mantid was fed a *B. anynana* with its eyespots rubbed off.

During each trial, the mantid was attracted into the arena using a mealworm. A single butterfly was then placed 5 cm away from the mantid (with a morsel of banana as an incentive to remain in place), facing perpendicular to the long axis of the ramp so as to maximally display its eyespots to the mantid. The prey–predator interaction that followed was recorded using a Legria HFR36 video camera. A new butterfly was used for each trial. If the butterfly flew away before an attack was initiated, the trial was repeated with another butterfly. A total of 10 mantids were used, and one round of trials comprised 10 trials: one per mantid in random sequence. After the first round, the mantids were again starved for two days before conducting the second round of trials. This process was repeated until eight rounds were completed. Each mantid encountered four Wt and four Spotty butterflies in random order. From the videos, we identified where on the butterfly the mantid first struck. Latency to attack was not recorded as mantids were observed to initiate attacks only in response to a movement by the butterflies. First strikes observed were assigned to one of four categories (figure 3*b*): (i) on or near the 'body'

(including the proximal parts of the wing, without any eyespots); (ii) on the 'forewing eyespots'; (iii) on the 'hindwing eyespots'; and (iv) on the 'eyespots on both wings' simultaneously.

The data were analysed in R (v. 4.0.0) [26]. A multinomial regression was performed using the nnet [32] and car [33] packages. An extension of binomial regression (which can only analyse two categories), multinomial regression compares the overall proportions of the four categories of first strikes observed to determine whether the mantids attacked Wt and Spotty differently. The response variable was the proportion of each first strike category and the explanatory variable was butterfly form. The multinomial model also compared all possible pairs of first strike categories to identify which specific categories were different. This procedure is similar to using an ANOVA to test for global differences among groups followed by individual *t*-tests to identify the groups driving the difference. To account for potential differences among the mantids, we used the lme4 package [34] to compare all pairwise combinations of the four categories via a series of binomial GLMMs, with form as the main effect and mantid identity as a random effect. This is the manual equivalent of the multinomial regression [35] but also allows the modelling of random effects. The arm package [36] was used for model checking.

# 3. Results

## (a) Wing overlap partially explains the uneven eyespot distribution

Of the 451 species from Tokita *et al.* [13], we excluded 20 species with no suitable images and 156 species without eyespots, leaving a total of 275 species for the three analyses. In the first analysis, we found 1.5 times more eyespots per exposed wing sector on hindwings (mean ± s.e.: 0.49 ± 0.015) than on forewings (0.31 ± 0.018). Second, wing overlap is a limiting factor for the number of forewing eyespots in only 35 species (12.8%). These species have fewer exposed forewing sectors than hindwing sectors with eyespots and therefore cannot have as many exposed forewing eyespots as hindwing eyespots. The remaining 240 species (87.2%), including *B. anynana*, have the potential to display the same number or more exposed forewing than hindwing eyespots and yet they have, on average, more than two times as many exposed eyespots on the hindwing (mean ± s.e.: 4.0 ± 0.14) than the forewing (1.8 ± 0.12), suggesting that this difference was not caused by wing overlap. Third, even though both the forewing M2 and M3 sectors are covered at rest in 13.8% (i.e. 38) of the 275 species, this is insufficient to explain why eyespots are twice as common on the homologous M2 and M3 hindwing sectors—present on the hindwing in 127 species (46.2%) but on the forewing in only 57 species (20.7%). Taken together, these three observations suggest that the lower eyespot number on forewings versus hindwings in nymphalids cannot be fully explained by wing overlap at rest.

## (b) Microcosm experiment: Spotty butterflies have lower fitness and suffer more wing damage

All 40 cages were used in the survival and fecundity analyses, although nine (two control Spotty, three control Wt and four experimental Wt cages) were censored at various points, e.g. owing to an escaped butterfly. Of these nine, six were censored from the wing damage location data, e.g. owing to premature mantid death.

There were three main results from the survival analysis (figure 2*a*). First, butterflies in experimental cages had shorter lifespans than those in control cages (6.7 versus 28.7 days; $p < 0.001$). Second, in control cages, butterfly form did not affect survival: there was no significant difference in the longevity of Wt (29.4 days) and Spotty (27.2 days) ($p = 0.48$). Third, in experimental cages, form significantly affected survival: Spotty (which survived an average of 3.4 days) died sooner than Wt (9.4 days; $p = 0.032$). Mantid gender was also significant (butterflies in cages with female mantids survived 2.5 days less; $p < 0.01$) but there was no significant interaction between mantid gender and butterfly form ($p = 0.48$).

The fecundity data produced similar results (figure 2*b*). First, butterflies in experimental cages laid half as many eggs as those in control cages (9.3 versus 18.9 eggs butterfly$^{-1}$ d$^{-1}$; $p < 0.001$). Second, in control cages, there was no significant difference between Wt and Spotty (20.3 versus 17.5 eggs butterfly$^{-1}$ d$^{-1}$, respectively; $p = 0.38$). Third, in experimental cages, form had a significant effect on fecundity: Spotty laid half as many eggs as Wt (6.8 versus 13.4 eggs butterfly$^{-1}$ d$^{-1}$; $p = 0.022$). This lower fecundity is probably driven by increased predation on Spotty.

There were also three main results from the wing damage data. First, butterflies in experimental cages suffered more damage than those in control cages on both forewings (0.36 damage marks butterfly$^{-1}$ d$^{-1}$ in experimental cages versus 0.16 marks in control cages; $p = 0.01$) and hindwings (0.28 versus 0.20 marks butterfly$^{-1}$ d$^{-1}$; $p = 0.01$) (electronic supplementary material, figure S2). Second, in control cages, there was no difference in wing damage between Wt and Spotty, both on forewings ($p = 0.80$) and hindwings ($p = 0.83$). Third, in experimental cages, Spotty were attacked more than Wt on both the forewing (0.22 damage marks butterfly$^{-1}$ d$^{-1}$ in Spotty versus 0.14 in Wt) and hindwing (0.17 versus 0.10 marks butterfly$^{-1}$ d$^{-1}$). The two-way average damage score heat map (figure 2*c*) shows this same pattern (the figure is predominantly yellow) and also suggests that the mantids tended to attack Spotty more on the anterior portions of both wings (where the darker yellows tend to occur). These differences border on significance for the forewing ($p = 0.083$; despite this *p*-value being greater than 0.05, the model including butterfly form explains the data significantly better than the model without it, $p = 0.04$, suggesting that the effect of form is significant) but not significant on the hindwing ($p = 0.19$). Together, these results indicate that Spotty experienced more intense predation and were attacked more on their forewings compared to Wt.

## (c) Arena experiment: Spotty and wild-type butterflies are targeted differently

Eight mantids successfully completed all eight trials. Two died prematurely, completing four trials (two Wt, two Spotty) and six trials (four Wt, two Spotty). A total of 69 attacks were recorded (35 on Wt and 34 on Spotty) and five trials (four Wt, one Spotty) resulted in no attack. In eight out of the 69 attacks observed (four Wt, four Spotty), the butterfly subsequently escaped. The number of first strikes in each category for Wt and Spotty is shown in table 1. As expected, most attacks targeted areas with eyespots (50 attacks, or 72%) (table 1). Focusing on the forewings: because the forewing is also targeted in attacks on both wings, the forewing was attacked more frequently in Spotty (38.2% on both wings + 29.4% on the forewing alone = 67.6% of trials) than in Wt (17.1% + 40.0% = 57.1% of trials) (figure 3*c*).

The multinomial GLMM revealed a difference which borders on significance in the overall proportions of the four first strike categories between Wt and Spotty ($p = 0.063$), supporting our wing damage data (figure 2*c*). The pairwise comparisons showed two main drivers of this difference (figure 3*c*). First, a significant difference in the relative proportion between 'hindwing eyespots' (reducing steeply from 17.1% in Wt to 2.9% in Spotty) and 'eyespots on both wings' (more than doubling from 17.1% in Wt to 38.2% in Spotty) ($p = 0.031$), suggesting that the mantids shifted from attacking the hindwings alone to attacking both wings simultaneously. Second, a difference bordering on significance between 'forewing eyespots' (decreasing from 40% in Wt to 29.4% in Spotty) and 'eyespots on both wings' ($p = 0.085$). All other comparisons were non-significant. The binomial GLMMs produced almost identical statistics, confirming these results, and indicated no significant difference among mantids. These data suggest that the predominant change in mantid attacks was a switch from targeting the hindwing in Wt to attacking both wings in Spotty.

**Table 1.** Total number of first strikes in each category on Spotty (Sp) and wild-type (Wt) butterflies. (Bo, body; FwE, forewing eyespots; HwE, hindwing eyespots; BE, eyespots on both wings.)

| mantid number | Bo (Wt) | Bo (Sp) | FwE (Wt) | FwE (Sp) | HwE (Wt) | HwE (Sp) | BE (Wt) | BE (Sp) | total |
|---|---|---|---|---|---|---|---|---|---|
| 1 | 2 | 1 | 2 | 0 | 0 | 0 | 0 | 2 | 7 |
| 2 | 1 | 2 | 1 | 0 | 0 | 0 | 1 | 2 | 7 |
| 3 | 1 | 1 | 2 | 3 | 0 | 0 | 0 | 0 | 7 |
| 4 | 1 | 1 | 2 | 1 | 0 | 0 | 1 | 2 | 8 |
| 5 | 1 | 2 | 3 | 0 | 0 | 1 | 0 | 0 | 7 |
| 6[a] | 1 | 0 | 1 | 2 | 1 | 0 | 1 | 0 | 6 |
| 7 | 0 | 2 | 2 | 1 | 1 | 0 | 1 | 1 | 8 |
| 8 | 0 | 0 | 0 | 1 | 2 | 0 | 2 | 3 | 8 |
| 9 | 0 | 0 | 1 | 2 | 2 | 0 | 0 | 2 | 7 |
| 10[a] | 2 | 1 | 0 | 0 | 0 | 0 | 0 | 1 | 4 |
| total | 9 | 10 | 14 | 10 | 6 | 1 | 6 | 13 | 69 |

[a]Mantid died before completing all eight trials.

## 4. Discussion

Multiple small marginal eyespots on hindwings are known to deflect predator attacks to the wing margin [6,7,11] but it is unclear why nymphalids generally have fewer eyespots on forewings than hindwings. Here, we find that this pattern on the ventral wing surfaces persists even after accounting for wing overlap in the butterflies' perching positions. We also demonstrate that *B. anynana* butterflies with more forewing eyespots suffered more attacks by mantid predators on the forewings, reducing their survival and fecundity. Our data indicate that predation helps to maintain lower forewing eyespot numbers in *B. anynana* because carrying more forewing eyespots draws attacks towards the forewing which is more important for flight.

Overall, our data show that both wing overlap and predation pressure contribute to limiting the number of ventral forewing eyespots in butterflies. However, wing overlap is a limiting factor in only 12.8% of the species analysed. Owing to the paucity of this type of data, it is likely that we did not capture all the resting positions displayed by these species in nature. However, because our analysis was deliberately conservative, we believe that, even as more images become available, our conclusions should remain sound. This does not immediately indicate that predation is primarily responsible for fewer forewing eyespots, but our data suggest that it might be an important factor. The presence of predators was detrimental to the fitness of both *B. anynana* forms in our study, but Spotty butterflies were significantly shorter-lived (figure 2a) and laid significantly fewer eggs (figure 2b). The mantids preyed upon the two forms differently: Spotty forewings both suffered more damage in the microcosm experiments (0.22 marks butterfly$^{-1}$ d$^{-1}$ versus 0.14 in Wt) and were attacked more frequently in the arena experiments (67.6% of Spotty trials versus 57.1% of Wt trials). While we did not explicitly control for wing damage owing to wear and tear in our microcosm experimental cages, our data show that this accumulates over time primarily on the leading edge of the forewing (electronic supplementary material, figure S2). Therefore, correcting for wear and tear would result in a greater reduction of damage scores on the forewings of Wt (which survived longer than Spotty), which would provide stronger support for our conclusions.

Although we show that having more forewing eyespots is detrimental, our results alone cannot disentangle whether this is a consequence of Spotty having a more even distribution of eyespots (across forewings and hindwings) or more eyespots overall. However, when informed by existing studies, our data suggest that eyespot location may be important. Prudic *et al.* [7] showed that, although the conspicuous wet season form (WS) of *B. anynana* was more attacked than the less conspicuous dry season form (DS), the WS still had higher fitness (because most damage was on the hindwings compared to the head and thorax in DS). Hence, if Spotty were attacked more, simply because they were more conspicuous (having more eyespots overall), we would expect more wing damage but not necessarily lower fitness. However, we observed both in the microcosm experiment. We therefore believe that the lower fitness in Spotty is owing to greater forewing relative to hindwing damage. In the arena experiment, the additional forewing eyespots in Spotty caused a significant reduction of attacks towards hindwings and an increase towards both wings, suggesting that the forewing became a relatively more important target. Nymphalids display large variation in total eyespot number but consistently have fewer forewing eyespots [13]. We believe our experiments shed light on the detrimental effect of drawing too much attention to forewings by placing too many eyespots on these wings, relative to placing most eyespots on hindwings. Nevertheless, future studies are needed to differentiate between the effects of total eyespot number and eyespot distribution, e.g. by painting eyespots onto the same butterfly genotype to produce forms with different eyespot distributions but the same total eyespot number.

Displaying more deflective eyespots on the hindwing appears to be beneficial as their outer margins are easily ripped and detached, allowing the prey to escape with only hindwing damage [37]. Butterfly hindwings are less important for flight: hindwing clippings are generally not

detrimental to survival (e.g. [38]); many species in the wild exhibit more hindwing damage [39]; and *Pieris rapae* and *Morpho* butterflies without hindwings remain able to evade predators (although at reduced height, speed and acceleration) [40,41]. Forewings, however, are critical for all stages of flight, from general flight to evasive manoeuvring [40–43]. A greater number of forewing eyespots, as seen in Spotty, may therefore be selectively disadvantageous because this encourages predators to grasp both wings simultaneously. This curtails a butterfly's chances of escaping an attack [44] and, if it does escape, still results in forewing damage, diminishing its survivability in subsequent encounters with predators.

The lower fecundity of Spotty relative to Wt in the presence of mantids could also be owing to lower mating success. In *Bicyclus* spp., including *B. anynana*, there is evidence that forewing pattern elements are more important for sexual signalling [17,45,46]. The greater damage to Spotty forewings could thus have contributed to lower mating frequency. However, in species which use both forewings and hindwings for sexual signalling, e.g. *Hypolimnas bolina* [47], the extent of wing damage may be more important than damage location *per se*. Few studies have investigated the relative importance of forewing versus hindwing markings in mating, and more experiments are necessary to evaluate how variation in eyespot distribution may affect sexual signalling in nymphalids. These studies should, however, consider possible sex-biased mortality (as sex ratio within a cage affects fecundity) and could also investigate female preference in the absence of harassment from males.

Finally, other factors could explain our results. Systematic differences in escape behaviour between Spotty and Wt, which we did not notice during our experiments, could result in different survival rates, although Prudic *et al*. [7] showed that the effects of eyespots on survival were independent of behaviour in the seasonal forms of *B. anynana*. Also, the M3 and Cu1 eyespots on the Spotty forewing are partially joined (the orange ring is always fused, and the inner black ring is fused in 10% of specimens), producing a new shape on the forewing which could affect predator behaviour. However, this would not affect the conspicuity of the white eyespot centres. They should therefore still draw attacks to themselves in a manner similar to isolated eyespots [12].

Nevertheless, future research should ensure that eyespot shape remains consistent. Startle displays, which have received limited study, could also potentially influence eyespot shape, size and distribution, and are an interesting avenue for further research. To extrapolate our conclusions to the Nymphalidae as a whole, similar work on other species is necessary. However, it should be considered that different predators may respond differently to the same wing pattern, and different predator guilds in different forests may explain why some nymphalids have more eyespots on the forewing instead. Hence, future work sampling whole predator–prey communities is needed to address further drivers of eyespot number diversity.

Ethics. All applicable guidelines from the Animal Behavior Society Guidelines for the Use of Animals in Research (Animal Behaviour, 2018, 135, I-X), national legal requirements and institutional guidelines were adhered to in conducting this study. A permit to collect mantid specimens was issued by the National Parks Board, Singapore (NP/RP15-063-1a).

Data accessibility. All data and the R code for the wing overlap, microcosm and arena experiments are available from the Dryad Digital Repository: https://doi.org/10.5061/dryad.h9w0vt4gw [48].

Authors' contributions. I.Z.W.C.: data curation, formal analysis, investigation, methodology, writing—original draft, writing—review and editing; Z.C.N.: conceptualization, investigation, methodology, writing—original draft; L.N.: conceptualization, investigation, methodology, writing—review and editing; Y.L.: conceptualization, investigation, methodology, writing—review and editing; V.G.: investigation, writing—review and editing; A.M.: conceptualization, funding acquisition, methodology, project administration, resources, supervision, writing—review and editing. All authors gave final approval for publication and agreed to be held accountable for the work performed therein.

Competing interests. The authors declare that they have no competing interests.

Funding. This work was supported by the National Research Foundation, Singapore under the Investigatorship Programme (grant no. NRF-NRFI05-2019-0006), the National Research Foundation, Singapore under the CRP programme (grant no. NRF-CRP20-2017-0001) and the Ministry of Education, Singapore's Academic Research Fund (grant no. R-154-000-602-112).

Acknowledgements. The authors would like to thank Aaron Teo for his advice on data analysis, and the editorial board member and anonymous referees for comments which substantively improved this paper.

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
