## [Peer Review File · Proceedings of the Royal Society B: Biological Sciences]

Review History

Decision letter (RSPB-2020-2044.R0)

01-Sep-2020

Dear Mr Chan:

Thank you for submitting your manuscript RSPB-2020-2044 entitled "Predation explains asymmetric eyespot distribution across wings of nymphalid butterflies" to Proceedings B.

All manuscripts are assessed by a specialist member of the Editorial Board, who decides whether the manuscript is suitable for Proceedings B.

Unfortunately, your manuscript has been rejected at this stage of the assessment process. Competition for space is currently extremely severe, and we receive many more manuscripts than we are able to publish. On this occasion it was felt that your manuscript was unlikely to be able to compete successfully for a space in the journal.

Please find below the specialist Board member's comments. I hope you may find these useful should you wish to submit your manuscript elsewhere.

Sincerely,
The Proceedings B Team
<mailto:proceedingsb@royalsociety.org>

Board Member

Comments to Author(s):

The manuscript under consideration is about eyespots on the wings of *Bicyclus* butterflies.

Eyespots potentially have several different functions (e.g. sexual selection, deflection of predator attacks from the body of the butterfly or predator intimidation). The authors use a very neat experimental microcosm setup with two naturally occurring forms of *Bicyclus* butterflies. The two forms differ in the numbers of eyespots on the ventral parts of their forewings. While stationary, butterflies hold their wings in a vertical position, displaying the undersides of their wings. These two forms of butterfly were presented to mantid predators (in microcosm cages and in more controlled experimental situations) and the subsequent attacks of the mantids were analysed. The authors found that the presence of the extra eyespots on the forewing resulted in an increased proportion of attacks on the forewing area of the butterfly. Butterflies with more spots were also attacked more frequently and the fecundity of these microcosm populations were lower than butterfly populations with fewer forewing spots. The results suggest that the position and number of spots influence predator behaviour and in particular what parts of the butterfly are attacked (I am not sure how novel this is).

I thought that the experimental design was extremely compelling and that the results looked very solid. My main criticism of this manuscript is aimed at the justification of the study (this was the most novel angle of the paper). A previous study (Tokita et al 2013) revealed that *Bicyclus* butterflies tend to have more spots on their hindwings than their forewings. This observation was used in the introduction as the primary justification of the study: experiments to explain the asymmetry of spots on butterfly wings. However, there are much simpler explanations of asymmetric spotting on butterfly hind- vs forewings. The most simple is that the hind and forewings overlap so that much less of the forewing is displayed when the butterfly is resting (the hindwing is placed over the forewing). Consequently, having spots on much of the ventral parts of the forewing would serve no function as they would not be visible. It would seem that Tokita et al (2013) did not take this into account when they unveiled this trend of spotting asymmetry. I am not convinced by the existence of this trend because there was no attempt to control for how much surface area of hind and forewings are exposed to visual observers.

I would suggest two possible solutions to this problem. The first is that the authors of this study, re-analyse spotting asymmetry as part of this manuscript and try to control for exposed wing area to see if the trend of asymmetry still exists. This would strengthen the link between putative spotting asymmetry and the experiments presented here. The second is that the authors could completely downplay the link between their study and patterns of spotting asymmetry. This may be a more risky solution because the authors point out that there have been plenty of previous studies on the effects of butterfly eyespots, and so without this link, it is unclear whether reviewers will still find the experiments novel enough. Without this link, the manuscript would potentially be more about attack deflection, however, the authors had no butterfly control without spots and so it may be hard to come to any real conclusion about how spots may deflect attacks from the body of the insect.

Below, I have two more minor comments:

L39: I think that the term "attack deflection" needs to be clearly defined at this point. I have assumed that it means that predator attacks are deflected from more important parts of the insect towards less important parts.

L241 (results): The authors find that more spots on the forewing increases attacks to the forewing, mortality and attack rates- these are compelling components of fitness, but true fitness is the number of offspring produced - here fecundity of populations is measured but not fecundity at an individual level. I wondered how a damaged forewing vs a damaged hindwing may affect an individual's reproductive success: While a damaged forewing may inhibit flight and reduce survival, it is unclear how wing damage may affect mate choice and mate acquisition. It could be

that hind wing patterns are more important for mate acquisition, or that butterflies with damaged wings (hind or fore) may be equally unsuccessful. Consequently, from a selection point of view, it may not matter how wing pattern affects survival if survivors are all equally reproductively compromised. Perhaps damaged females may still easily obtain mates but damaged males may battle (this may mean that spotting is more important through the female fitness avenue). All of this may mean that rate of attack or presence of damage may be much more important than what parts were attacked. This may warrant more in-depth discussion.

RSPB-2020-2840.R0

Review form: Reviewer 1 (Carlos Cordero)

Recommendation

Major revision is needed (please make suggestions in comments)

Scientific importance: Is the manuscript an original and important contribution to its field?

Good

General interest: Is the paper of sufficient general interest?

Good

Quality of the paper: Is the overall quality of the paper suitable?

Good

Is the length of the paper justified?

Yes

Should the paper be seen by a specialist statistical reviewer?

Yes

Do you have any concerns about statistical analyses in this paper? If so, please specify them explicitly in your report.

No

It is a condition of publication that authors make their supporting data, code and materials available - either as supplementary material or hosted in an external repository. Please rate, if applicable, the supporting data on the following criteria.

Is it accessible?

Yes

Is it clear?

Yes

Is it adequate?

Yes

Do you have any ethical concerns with this paper?

No

Comments to the Author

Review - Predation explains asymmetric eyespot distribution across wings of nymphalid butterflies (RSPB-2020-2840)

This is a very interesting paper. The main aim was to contrast two hypotheses to explain the evolutionary maintenance of asymmetric eyespot distribution between fore- and hindwings. One hypothesis involves constraints and the other natural selection derived from visually oriented predators. The results from a comparative study and two experiments support the last hypothesis.

My main concerns with this manuscript are related to the Methods. In particular, more information is needed to understand how the comparative study and the arena experiment were done and, thus, to judge its results and interpretation. More general, I consider that these two studies cannot be replicated with the information provided in the manuscript.

First, with respect to the study comparing the part of the forewings that is covered by the hindwings (when perching) using photographs taken from different public repositories, I wonder how is it possible to determine what this area is for the whole species and why these estimates are presented as error free. I have experience observing perching butterflies in the field and recently, for reasons that have nothing to do with the subject of this manuscript, with one of my colleagues, I have been looking at photographs of different species of butterflies (not nymphalids, I must admit) that perch (and feed and oviposit) with their wings closed and I can attest that the front wing area covered by the hindwings varies within species (frequently with time in the same individual), and sometimes varies a lot. I think that it is necessary to explain clearly how these data were obtained, how much intraspecific variation was observed, the number of photographs on which the estimates are based, etc., to allow the reader to judge the results presented.

Second, in the arena experiment it is mentioned that encounters between one mantis and one butterfly were staged and that a total of 10 mantids were used, each was exposed in a random sequence to 8 butterflies (4 wild type and 4 spotty). To replicate this experiment it is needed to explain how was this done. For example, one mantis was exposed to the complete sequence of butterflies the same day or to how many per day? Different mantids were exposed to butterflies at the same time or there was only one arena and the different mantises were used in alternate moments? etc.

Finally, in lines 131-132 it is mentioned that only female mantids were used in the experiments, although this is clearly not the case in the "microcosm experiment" (mantis sex was used as an effect in the analyses). In the "arena experiment" the sex of the mantids is not mentioned and it is not included in the analyses. It is possible that it was in this experiment that only females were used.

Regarding the Results section:

In the results relative to the microcosm experiment, butterflies with different number of wing spots are referred to as "forms". I think this can be confusing, as for example when it is said that "in control cages, butterfly form did not affect survival".

In lines 274-277 the authors suggest that the difference in "fecundity" observed between Wt and Spotty butterflies is "likely driven by the shorter lifespan" of the Spotty butterflies. However, the data they present show differences in the number of eggs laid per day and I think that Figure 2b (lower panel) also does not support their interpretation. The reduced egg-laying rate might be a product of more mantis "harassment" on Spotty butterflies.

Finally, I wonder what proportion of attacks in the arena experiment resulted in predation and if there was a difference in this proportion between Wt and Spotty butterflies.

With respect to the Discussion:

In lines 363-365 I found two confusing assertions:

(1) “[M]antids predominantly shifted from targeting the hind wing only in Wt to targeting both wings in Spotty” – This assertion gives the impression that mantids attack predominantly the hindwings of Wt butterflies, which is not the case.

(2) “As the hindwing is targeted in both the forms, there would be a bigger difference on the forewings and a smaller difference (if any) on the hindwings” – I found this confusing because mantids barely directed attacks to the hind wing in Spotty butterflies (figure 3c), although they directed the larger proportion of their attacks to the eyespots of both wings. Also, Wt butterflies were most attacked in the eyespots of the forewing area. I feel that the confusion will disappear if the drafting of these assertions is improved.

In conclusion, although I have doubts about the comparative study, I think that the results of the two experiments support the hypothesis that selective pressures derived from predators are at least partially responsible for the evolutionary maintenance of asymmetric eyespot distribution between fore- and hindwings and deserve to be published, although more details of the experimental procedures need to be provided.

Review form: Reviewer 2

Recommendation

Reject – article is scientifically unsound

Scientific importance: Is the manuscript an original and important contribution to its field?

Good

General interest: Is the paper of sufficient general interest?

Good

Quality of the paper: Is the overall quality of the paper suitable?

Marginal

Is the length of the paper justified?

Yes

Should the paper be seen by a specialist statistical reviewer?

Yes

Do you have any concerns about statistical analyses in this paper? If so, please specify them explicitly in your report.

No

It is a condition of publication that authors make their supporting data, code and materials available - either as supplementary material or hosted in an external repository. Please rate, if applicable, the supporting data on the following criteria.

Is it accessible?

Yes

Is it clear?

Yes

Is it adequate?

Yes

Do you have any ethical concerns with this paper?

No

Comments to the Author

This is an interesting study. It was a pleasure to read the manuscript, and I think the analyses are well done. However, I have a couple of concerns regarding the conclusions drawn.

This sentence perhaps encapsulates what the authors infer as the main conclusion of the study: "Our data indicate that the location of eyespots on butterfly wings is important and suggest that predation is the primary driver behind the asymmetric eyespot distribution observed in *B. anynana* and most other nymphalid species.". The authors aim to explain why spots are more frequent on the hindwings compared to the forewings.

The study has two components. The first tests whether forewings have fewer eyespots than do hindwings simply because part of the forewing is hidden under the hindwing, and, therefore, there is an asymmetry in the area available to display to predators. Their analysis of spot numbers on the fore- and hindwings of butterfly species indicate that the asymmetry in eyespot numbers between the forewings and hindwings is not explained by space constraints.

The second component includes laboratory predation trials. Here, individuals with a mutant phenotype 'spotty' were pitted against wildtype butterflies. Compared to wild type ones, spotty butterflies had 2 additional eyespots on the forewing. The authors explicitly assume that there are no other differences between the two morphs apart from the 2 eyespots (L 83). Spotty butterflies had lower fitness as measured by multiple fitness proxies (e.g. how long it took the predator to consume the butterfly, fecundity, longevity, etc.). My first concern here is about how well controlled (or not) the experiments were. Is the presence of the two eyespots the ONLY difference between the two morphs? a) Looking at Fig 1, spotty individuals seem to have eyespots with larger white centres. b) Furthermore, the eyespots in Cu1 and M3 appear to be fused together, although such a fusion of adjacent spots does not seem to be the case in the hindwing. I did an online search for Spotty and found a couple of other papers (in which the last author of the current study was also involved in) with images of this mutant. It appears that the fusion of adjacent eyespots is found in other spotty individuals. Of course, I cannot and should not draw any conclusions based on a few images. But, we cannot also exclude the possibility that spotty and wildtype butterflies differ in aspects other than $n/n+2$ eyespots. For e.g., the authors themselves discuss (L 51) *Parage aegeria*, a butterfly with eyespot number polymorphism, where there are behavioral differences between the morphs. The fusion of adjacent eyespots could create a new shape (as opposed to a circle) that may have fitness consequences. Therefore, I am not convinced that the differences between spotty and wild type can be attributed confidently only to the effect of the two eyespots.

Furthermore, even if the difference in survival between the two morphs is explained solely by the additional two eyespots, I am not convinced that these results by themselves explain the broader pattern of forewing-hindwing asymmetry in butterflies. There could be a threshold effect in this particular species, wherein two additional eyespots are too many. In fact, this argument is bolstered by this statement in the Introduction "Furthermore, paper models of *B. anynana* with a greater number of small ventral marginal eyespots are more readily attacked, presumably because they are more conspicuous and attract more predator attention [6,12]". So, it is already known from previous work that too many eyespots on the ventral surface is not good (although having a few is better than having none).

To explain why eyespots are relatively infrequent on the forewing, a more appropriate experiment would be one involving a comparison between two treatments: 1) more eyespots on the forewing than on the hindwing 2) more eyespots on the hindwing than on the forewing. If the eyespot morphology is properly controlled, and fitness of individuals in the first treatment is higher, we would have stronger support for their hypothesis.

Furthermore, deflective eyespots have been shown to function by attracting attacks towards the wing margin, such that a butterfly can escape with a torn wing. Forewings are more important than hindwings for flight and mating success (see L404). Thus, a torn forewing could be more detrimental than a torn hindwing in terms of flight performance and mating success. Thus, the asymmetry in eyespot frequency between forewings and hindwings may be explained by the indirect (through wing damage) effects of BOTH predation AND mate choice. This is something that the authors have not tested here, but should probably discuss in the Discussion.

Some other comments

L39: Eyespots, bullseye patterns consisting of concentric rings of contrasting colours, are a common protective pattern in many lepidopteran species [3,4].

Please rephrase this. Eyespots are not always composed of concentric rings. In fact, many species in which eyespots have been studied - *Inachis io*, *Junonia* spp, etc - do not have eyespots with well defined concentric rings.

L115 Second, we compared the number of exposed forewing versus hindwing eyespots in the species where the number of exposed forewing sectors was not a limiting factor. We defined this number as a limiting factor if it is less than the number of eyespots on the hindwing, i.e. even if all exposed forewing sectors bore an eyespot, there would still be fewer exposed eyespots on the forewing than the hindwing.

This was one of the few places where I had to read more than once, in order to try and understand. This was hard to follow - please consider rephrasing this.

L 274. "Spotty butterflies laid half as many as eggs as Wt butterflies over the course of their lives (6.8 versus

276 13.4 eggs per butterfly per day; $P = 0.022$). This lower fecundity is likely driven by the shorter life span of Spotty animals relative to Wt animals."

This was perplexing to me. You measured the number of eggs laid per day. I presume you only counted the number of days a butterfly lived. If so, why should longevity affect the number of eggs laid per day? In fact, some animals live shorter but have higher per-day fecundity.

Decision letter (RSPB-2020-2840.R0)

22-Dec-2020

Dear Dr Chan:

Your manuscript has now been peer reviewed and the reviews have been assessed by an Associate Editor. The reviewers' comments (not including confidential comments to the Editor) and the comments from the Associate Editor are included at the end of this email for your reference. As you will see, the reviewers and the Editors have raised some concerns with your manuscript and we would like to invite you to revise your manuscript to address them.

Research ethics:

Use of animals and field studies:

It is a condition of publication that you make available the data and research materials supporting the results in the article (<https://royalsociety.org/journals/authors/author-guidelines/#data>). Datasets should be deposited in an appropriate publicly available repository and details of the associated accession number, link or DOI to the datasets must be included in the Data Accessibility section of the article (<https://royalsociety.org/journals/ethics-policies/data-sharing-mining/>). Reference(s) to datasets should also be included in the reference list of the article with DOIs (where available).

All supplementary materials accompanying an accepted article will be treated as in their final form. They will be published alongside the paper on the journal website and posted on the online

figshare repository. Files on figshare will be made available approximately one week before the accompanying article so that the supplementary material can be attributed a unique DOI. Please try to submit all supplementary material as a single file.

Please submit a copy of your revised paper within three weeks. If we do not hear from you within this time your manuscript will be rejected. If you are unable to meet this deadline please let us know as soon as possible, as we may be able to grant a short extension.

Best wishes,
Dr Locke Rowe
mailto:proceedingsb@royalsociety.org

Associate Editor Board Member

Comments to Author:

This is the second time I am handling this manuscript. The first time, I read the manuscript thoroughly and suggested multiple additions and changes before the manuscript was sent for review. These changes were made by the authors and the manuscript was sent to two reviewers. I now report back on the findings of the two reviewers, both of whom thought that the manuscript was interesting. Despite the general interest by both reviewers, both also have some critical comments for the authors to address. The first reviewer outlines various places in the methods where crucial information is missing, making it impossible for the experiments to be accurately replicated. I implore the authors to address these comments carefully and include the relevant information. Among other points, the second reviewer writes that it is possible that spot number is not the only difference between the two butterfly forms and so it is possible that the observed results are not due to differences in spot number. Here I suggest that some of the points brought up by reviewer 2 are critically discussed in the discussion. This reviewer also wrote to me that "it would be good for a specialist statistical reviewer to check whether the authors have appropriately taken into account multiple measurements from individual mantids in the 'Arena' experiment (the whole experiment only included 10 mantids in total)." I realize that the authors have used mantid ID as a random factor in their models, but a response to this point will also help me decide on how to proceed with the manuscript.

Please respond carefully to all of the reviewer comments.

Reviewer(s)' Comments to Author:

Referee: 1

Comments to the Author(s).

Review - Predation explains asymmetric eyespot distribution across wings of nymphalid butterflies (RSPB-2020-2840)

This is a very interesting paper. The main aim was to contrast two hypotheses to explain the evolutionary maintenance of asymmetric eyespot distribution between fore- and hindwings. One hypothesis involves constraints and the other natural selection derived from visually oriented predators. The results from a comparative study and two experiments support the last hypothesis.

My main concerns with this manuscript are related to the Methods. In particular, more information is needed to understand how the comparative study and the arena experiment were done and, thus, to judge its results and interpretation. More general, I consider that these two studies cannot be replicated with the information provided in the manuscript.

First, with respect to the study comparing the part of the forewings that is covered by the hindwings (when perching) using photographs taken from different public repositories, I wonder how is it possible to determine what this area is for the whole species and why these estimates are presented as error free. I have experience observing perching butterflies in the field and recently, for reasons that have nothing to do with the subject of this manuscript, with one of my colleagues, I have been looking at photographs of different species of butterflies (not nymphalids, I must admit) that perch (and feed and oviposit) with their wings closed and I can attest that the front wing area covered by the hindwings varies within species (frequently with time in the same individual), and sometimes varies a lot. I think that it is necessary to explain clearly how these data were obtained, how much intraspecific variation was observed, the number of photographs on which the estimates are based, etc., to allow the reader to judge the results presented.

Second, in the arena experiment it is mentioned that encounters between one mantis and one butterfly were staged and that a total of 10 mantids were used, each was exposed in a random sequence to 8 butterflies (4 wild type and 4 spotty). To replicate this experiment it is needed to explain how was this done. For example, one mantis was exposed to the complete sequence of butterflies the same day or to how many per day? Different mantids were exposed to butterflies at the same time or there was only one arena and the different mantises were used in alternate moments? etc.

Finally, in lines 131-132 it is mentioned that only female mantids were used in the experiments, although this is clearly not the case in the "microcosm experiment" (mantis sex was used as an effect in the analyses). In the "arena experiment" the sex of the mantids is not mentioned and it is not included in the analyses. It is possible that it was in this experiment that only females were used.

Regarding the Results section:

In the results relative to the microcosm experiment, butterflies with different number of wing spots are referred to as "forms". I think this can be confusing, as for example when it is said that "in control cages, butterfly form did not affect survival".

In lines 274-277 the authors suggest that the difference in "fecundity" observed between Wt and Spotty butterflies is "likely driven by the shorter lifespan" of the Spotty butterflies. However, the data they present show differences in the number of eggs laid per day and I think that Figure 2b (lower panel) also does not support their interpretation. The reduced egg-laying rate might be a product of more mantis "harassment" on Spotty butterflies.

Finally, I wonder what proportion of attacks in the arena experiment resulted in predation and if there was a difference in this proportion between Wt and Spotty butterflies.

With respect to the Discussion:

In lines 363-365 I found two confusing assertions:

- (1) "[M]antids predominantly shifted from targeting the hind wing only in Wt to targeting both wings in Spotty" - This assertion gives the impression that mantids attack predominantly the hindwings of Wt butterflies, which is not the case.
- (2) "As the hindwing is targeted in both the forms, there would be a bigger difference on the forewings and a smaller difference (if any) on the hindwings" - I found this confusing because mantids barely directed attacks to the hind wing in Spotty butterflies (figure 3c), although they directed the larger proportion of their attacks to the eyespots of both wings. Also, Wt butterflies

were most attacked in the eyespots of the forewing area. I feel that the confusion will disappear if the drafting of these assertions is improved.

In conclusion, although I have doubts about the comparative study, I think that the results of the two experiments support the hypothesis that selective pressures derived from predators are at least partially responsible for the evolutionary maintenance of asymmetric eyespot distribution between fore- and hindwings and deserve to be published, although more details of the experimental procedures need to be provided.

Referee: 2

Comments to the Author(s).

This is an interesting study. It was a pleasure to read the manuscript, and I think the analyses are well done. However, I have a couple of concerns regarding the conclusions drawn.

This sentence perhaps encapsulates what the authors infer as the main conclusion of the study: "Our data indicate that the location of eyespots on butterfly wings is important and suggest that predation is the primary driver behind the asymmetric eyespot distribution observed in *B. anynana* and most other nymphalid species.". The authors aim to explain why spots are more frequent on the hindwings compared to the forewings.

The study has two components. The first tests whether forewings have fewer eyespots than do hindwings simply because part of the forewing is hidden under the hindwing, and, therefore, there is an asymmetry in the area available to display to predators. Their analysis of spot numbers on the fore- and hindwings of butterfly species indicate that the asymmetry in eyespot numbers between the forewings and hindwings is not explained by space constraints.

The second component includes laboratory predation trials. Here, individuals with a mutant phenotype 'spotty' were pitted against wildtype butterflies. Compared to wild type ones, spotty butterflies had 2 additional eyespots on the forewing. The authors explicitly assume that there are no other differences between the two morphs apart from the 2 eyespots (L 83). Spotty butterflies had lower fitness as measured by multiple fitness proxies (e.g. how long it took the predator to consume the butterfly, fecundity, longevity, etc.). My first concern here is about how well controlled (or not) the experiments were. Is the presence of the two eyespots the ONLY difference between the two morphs? a) Looking at Fig 1, spotty individuals seem to have eyespots with larger white centres. b) Furthermore, the eyespots in Cu1 and M3 appear to be fused together, although such a fusion of adjacent spots does not seem to be the case in the hindwing. I did an online search for Spotty and found a couple of other papers (in which the last author of the current study was also involved in) with images of this mutant. It appears that the fusion of adjacent eyespots is found in other spotty individuals. Of course, I cannot and should not draw any conclusions based on a few images. But, we cannot also exclude the possibility that spotty and wildtype butterflies differ in aspects other than $n/n+2$ eyespots. For e.g., the authors themselves discuss (L 51) *Parage aegeria*, a butterfly with eyespot number polymorphism, where there are behavioral differences between the morphs. The fusion of adjacent eyespots could create a new shape (as opposed to a circle) that may have fitness consequences. Therefore, I am not convinced that the differences between spotty and wild type can be attributed confidently only to the effect of the two eyespots.

Furthermore, even if the difference in survival between the two morphs is explained solely by the additional two eyespots, I am not convinced that these results by themselves explain the broader pattern of forewing-hindwing asymmetry in butterflies. There could be a threshold effect in this particular species, wherein two additional eyespots are too many. In fact, this argument is bolstered by this statement in the Introduction "Furthermore, paper models of *B. anynana* with a greater number of small ventral marginal eyespots are more readily attacked, presumably because they are more conspicuous and attract more predator attention [6,12]". So, it is already

known from previous work that too many eyespots on the ventral surface is not good (although having a few is better than having none).

To explain why eyespots are relatively infrequent on the forewing, a more appropriate experiment would be one involving a comparison between two treatments: 1) more eyespots on the forewing than on the hindwing 2) more eyespots on the hindwing than on the forewing. If the eyespot morphology is properly controlled, and fitness of individuals in the first treatment is higher, we would have stronger support for their hypothesis.

Furthermore, deflective eyespots have been shown to function by attracting attacks towards the wing margin, such that a butterfly can escape with a torn wing. Forewings are more important than hindwings for flight and mating success (see L404). Thus, a torn forewing could be more detrimental than a torn hindwing in terms of flight performance and mating success. Thus, the asymmetry in eyespot frequency between forewings and hindwings may be explained by the indirect (through wing damage) effects of BOTH predation AND mate choice. This is something that the authors have not tested here, but should probably discuss in the Discussion.

Some other comments

L39: Eyespots, bullseye patterns consisting of concentric rings of contrasting colours, are a common protective pattern in many lepidopteran species [3,4].

Please rephrase this. Eyespots are not always composed of concentric rings. In fact, many species in which eyespots have been studied -*Inachis io*, *Junonia* spp, etc - do not have eyespots with well defined concentric rings.

L115 Second, we compared the number of exposed forewing versus hindwing eyespots in the species where the number of exposed forewing sectors was not a limiting factor. We defined this number as a limiting factor if it is less than the number of eyespots on the hindwing, i.e. even if all exposed forewing sectors bore an eyespot, there would still be fewer exposed eyespots on the forewing than the hindwing.

This was one of the few places where I had to read more than once, in order to try and understand. This was hard to follow - please consider rephrasing this.

L 274. "Spotty butterflies laid half as many as eggs as Wt butterflies over the course of their lives (6.8 versus

276 13.4 eggs per butterfly per day; $P = 0.022$). This lower fecundity is likely driven by the shorter life span of Spotty animals relative to Wt animals."

This was perplexing to me. You measured the number of eggs laid per day. I presume you only counted the number of days a butterfly lived. If so, why should longevity affect the number of eggs laid per day? In fact, some animals live shorter but have higher per-day fecundity.

Author's Response to Decision Letter for (RSPB-2020-2840.R0)

See Appendix A.

RSPB-2020-2840.R1 (Revision)

Review form: Reviewer 1 (Carlos Cordero)

Recommendation

Accept as is

Scientific importance: Is the manuscript an original and important contribution to its field?

Good

General interest: Is the paper of sufficient general interest?

Good

Quality of the paper: Is the overall quality of the paper suitable?

Excellent

Is the length of the paper justified?

Yes

Should the paper be seen by a specialist statistical reviewer?

No

Do you have any concerns about statistical analyses in this paper? If so, please specify them explicitly in your report.

No

It is a condition of publication that authors make their supporting data, code and materials available - either as supplementary material or hosted in an external repository. Please rate, if applicable, the supporting data on the following criteria.

Is it accessible?

Yes

Is it clear?

Yes

Is it adequate?

Yes

Do you have any ethical concerns with this paper?

No

Comments to the Author

Congratulations for your interesting paper.

Review form: Reviewer 3

Recommendation

Accept with minor revision (please list in comments)

Scientific importance: Is the manuscript an original and important contribution to its field?

Acceptable

General interest: Is the paper of sufficient general interest?

Marginal

Quality of the paper: Is the overall quality of the paper suitable?

Acceptable

Is the length of the paper justified?

Yes

Should the paper be seen by a specialist statistical reviewer?

No

Do you have any concerns about statistical analyses in this paper? If so, please specify them explicitly in your report.

No

It is a condition of publication that authors make their supporting data, code and materials available - either as supplementary material or hosted in an external repository. Please rate, if applicable, the supporting data on the following criteria.

Is it accessible?

Yes

Is it clear?

Yes

Is it adequate?

Yes

Do you have any ethical concerns with this paper?

No

Comments to the Author

The author presents a very well written manuscript on the asymmetric distribution of eyespots in nymphalid butterfly wings. The work is embedded in relevant literature and the manuscript is nicely structured and well-organized. Their results of a comparative study and two experiments provide support for the hypothesis that selective predation plays a role in the evolution and maintenance of eyespot asymmetry (between fore- and hindwings). I have read the current version of the manuscript in detail and gone over the authors responses to reviewers comments. I believe that the authors have satisfactorily addressed all comments. I agree that the results of this study are interesting and important, and they deserve to be published.

Having said this, I think statements such as 'predation explains' in the title, and 'predation is the primary driver behind the asymmetric eyespot distribution observed in *B. anynana* and most other nymphalid species' in the Discussion are a not justified by the data and should be tuned-down. The work indeed reveals that predation may play a role (and this is a valuable contribution in its own right), but more data is needed (e.g. confirmation from other species and other predators, etc) to validate that predation is in fact the 'primary driver'.

As a more general comment, I believe that the comparative study as conducted here, using few images from field guides, excluding intraspecific variation and phylogenetic effects, is far from ideal. Nymphalidae demonstrate a plethora of resting behaviours, and even within a single *Mycalesina* species there is ample variation in the relative exposure of the forewing (and even the display of dorsal vs ventral). *B. anynana*, for example, only 'folds away' their forewing after resting for a while (and this state will not often be presented in field guides because it is less

informative from a taxonomical perspective). Nevertheless, I realize these data are very hard to come by, and I am confident that the observed pattern (i.e. wing overlap is a not a limiting factor) will hold.

Minor comments:

For the arena experiment, giant Asian mantis nymphs *Hierodula* sp. were trapped IN a park along Upper Serangoon Road, Singapore. THEY were reared to adulthood [in the laboratory] AND only female mantids were used.

\Erik van Bergen

Review form: Reviewer 4

Recommendation

Major revision is needed (please make suggestions in comments)

Scientific importance: Is the manuscript an original and important contribution to its field?

Good

General interest: Is the paper of sufficient general interest?

Good

Quality of the paper: Is the overall quality of the paper suitable?

Good

Is the length of the paper justified?

Yes

Should the paper be seen by a specialist statistical reviewer?

No

Do you have any concerns about statistical analyses in this paper? If so, please specify them explicitly in your report.

No

It is a condition of publication that authors make their supporting data, code and materials available - either as supplementary material or hosted in an external repository. Please rate, if applicable, the supporting data on the following criteria.

Is it accessible?

N/A

Is it clear?

N/A

Is it adequate?

N/A

Do you have any ethical concerns with this paper?

No

Comments to the Author

This manuscript aims at exploring the evolutionary forces involved in the uneven number of eyespots in the forewing vs. hindwing observed in many Nymphalid butterflies. First, the degree of wing overlap on the ventral forewing pattern of 275 species is explored and the authors conclude that the reduced number of eyespots on the ventral side of the forewing probably not rely on the lack of positive selection due to wing overlap preventing selection by visual predators.

The manuscript then also described two interesting experiments with mantid predators, comparing the wing areas attacked, as well as the fitness (survival and fecundity) of *B. anynana* butterflies differing in the number of forewing eyespots. Butterflies with increased number of eyespots (spotty mutants) suffers from increased attack rate and reduced fecundity, but the experiments exploring the mantid behaviour reveals that the increase number of eyespots in the forewing does not result in increased number of attacks in the forewing but rather in an increased attack number on both wings. These experiments thus do not directly provide a selective explanation for the evolution of uneven number of eyespots in the ventral side of forewing vs. hindwing. The main caveats being that the experiments rely on the comparison of two colour-patterns differing not only in the FW/HW distribution of eyespots but also in their total number of eyespots.

Nevertheless, the reported experiments bring new insights on the predation induced by eyespots located on the ventral side of butterflies, I would thus recommend bringing clarifications on the hypotheses tested and general conclusions brought by the experiments.

Specific comments:

Title and throughout the text: the term 'asymmetry' to refer to difference between the forewing and the hindwing is misleading. Asymmetry would rather be appropriate to compare right and left wings. I would recommend using more precise term such as 'uneven eyespots distribution between fore and hindwings'.

Abstract

Line 17: 'Eyespots are anti-predator defence in many butterflies': this general statement is misleading, although the eyespots have been shown to deflect attacks in some cases, the effect of eyespots on predator behaviour is not straightforward and remains to be investigated, and this study contributes to this understanding, I would avoid such over-general and vague sentence and rather start with the evolutionary question investigated in this study.

Line 17-18: It is also misleading to report the result of a previous study in the abstract.

Line 22: 275 nymphalid species ALL EXHIBITING EYESPOTS

Line 23: 'does not limit' is unclear. Did you mean that covered areas do display eyespots in many species? Here, and throughout the manuscript, it is a bit unclear why and how would the coverage of the forewing by the hindwing would prevent the evolution of eyespots? Is it because, those partially covered eyespot are less likely to be seen by predators, and therefore less likely to be under positive selection? This needs to be more explicit in the abstract and in the introduction.

Line 24: The rationale behind the two experiments should be provided here and in the introduction as well, one experiment allows to compare the fitness associated with spotty vs. wild types phenotype, while the other experiment focuses on predator behaviour and the targeted regions of wings.

Line 28-30: This interpretation of the results is not entirely correct: butterflies with increased number of eyespots (spotty mutants) suffers from increased attack rate and reduced fecundity, but the experiments exploring the mantid behaviour reveals that the increase number of eyespots in the forewing does not result in increased number of attacks in the forewing but rather in an increased attack number on both wings. These experiments thus do not directly provide a selective explanation for the evolution of uneven number of eyespots in the ventral side of forewing vs. hindwing. The main caveats being that the experiments rely on the comparison of two colour-patterns differing not only in the FW/HW distribution of eyespots but also in their total number of eyespots. The experiments clearly show that individuals with more eyespots suffers from increase attacks, but does not allow to conclude about the selection of uneven

number in FW/HW.

Introduction:

Line 40: the term 'function' sounds finalistic and is often misleading, because most trait evolve under multiple selective pressures. The evolution of eyespots can result from trade-offs between conspicuousness and deflection for instance, so that this sentence should be entirely rephrased to explain the selective pressures at play and highlight that their relative contribution remains to be studied.

Line 47: replace 'the eyespots' attack deflection function' by 'deflection of the attacks on the eyespots, located in less-vital parts of the body'.

Line 58: removed 'it is interesting that'.

You may also detail why you focus on the ventral side of the wings and do not investigate the effect coverage of wing on the evolution of eyespots on the dorsal side as well.

Line 63: replace 'be able to function as a continuously-visible signal' by 'be continuously-visible'

Line 64: the term 'useful' is definitely finalistic, and this misleading sentence prevent a proper description of the selective force involved: did you mean prevent positive selection due to attack deflection? The deimatic effect, although poorly studied should probably not be neglected, and is likely to depend on the relative size and shape of the different eyespots, that has not been investigated here. It is particularly surprising to discard this hypothesis so quickly, because as shown on fig.1A, the biggest and most conspicuous eyespot observed on the ventral side of the wings of *B. anynana* is located on the forewing, and in a partially covered area.

Line 67 and 73: A mentioned above, the term 'asymmetry' is not entirely appropriate here.

Line 92-95: The questions investigated in the two experiments should be more explicit: the microcosm experiments allow to estimate the effect of the two phenotypes on the fitness of the individuals by comparing survival and reproductive success with and without predators, while the arena experiments precisely investigates the region of the wings targeted by predator's attacks, and therefore the selection acting on the distribution of eyespots.

I would also advise to present first the behavioural experiment and then the impact on fitness.

Materials and Methods

Line 104-105: Is there a reason for ignoring behaviour variation within species? The mean position covered could be used? Would it change the results or is this variation observed for a large or a small proportion of the species? These data could be of interests for a large number of entomologists, the authors indeed provide a Dryad number, but I was not able to access it and therefore could not check them (maybe because the data will be online only after the publication?).

Line 123-125: Is there any reason to use different predators in the two experiments? And to use female only in the arena experiments while using both sex in the microcosm. As mentioned in the introduction, both are probably relevant predators, but it makes the two experiments less related to each other.

Line 126-127: Did the wild-type and mutant shared the same genetic background? Are spotty butterflies related to each other? An increase relatedness among individuals could affect their fertility within cage.

Line 143-144: Was wing damage quantified in both control and experimental cage? Did the butterflies have wing damage in cage without predators (because they rub their wing against the cage walls for instance). I guess Fig. 2C reports damage in cages with predators? It would be important to disentangle the effect of the damages cause by predators from the damages due to aging and rubbing against the cages' walls.

Line 147-148: The rationale behind the use of the 0.5 scoring is not entirely clear. Here, it is hard to assess whether a mark spreading two wing cells was due to a single attack or a succession of several attack and subsequent wing rubbing. It would probably be more conservative, to have a simpler binomial score for each wing cell describing whether it was intact or damage. For a quantitative measure, you would need to compute, for each cell, the proportion of the wing cell surface removed.

Line 166: Was the number of eggs laid divided by the number of butterflies alive during each day? Wouldn't it be more relevant to control for the number of females still alive, because only females are able to lay eggs? I am not familiar with *B. anynana*, but for other butterfly species, females do not remain virgin in a cage with males, and the number of eggs will thus strongly

depend on the sex-ratio, females being frequently harassed by males. It would be relevant to control for the changes in sex-ratio throughout the experiments when studying the number of eggs produced.

Line 167: 'problems with error distribution', this must be more precise.

Line 171: the percentages reported on the heat map (fig.2C) are not straightforward: for example, the green part on the leading edge of the FW: does it mean that twice as much wt individuals were damaged at this location with respect to spotty individuals? Is it standardized by the total number of individuals damaged at this location? Please provide more detail on the way these percentages were computed.

Line 188 and line 131: the general purpose of each of the two experiments should be given at the beginning of the sections, for the reader to immediately know what will be tested in each experiment.

Line 216: I did not understand why you studied the first four strikes while each mantid was tested 8 times (with four spotty and four wt butterflies, as mentioned on line 208).

Line 217: Does the 'type of strike' relate to the position targeted by the mantid (as described on line 296-300)? If so, this should be clearly stated here.

Line 219: Are 'first strike categories' similar to the 'type of first strike'? It would be easier to define those types or categories and use a consistent vocabulary.

Results

Line 235: 'Wing overlap is a limiting factor': this is an interpretation, Does it mean that in 12.8% of species, no eyespot were observed in the FW zone covered by the HW? This should be precisely described. Similarly, on line 238, it is rather difficult to understand what is the result for the 240 remaining species? Do they display eyespot in the FW zone covered by the HW?

If I understood correctly, only a limited number of species had no eyespots in the covered part of the forewing despite displaying eyespots on the rest of the wings. The authors conclude that the reduced number of eyespots on the ventral side of the forewing probably not rely on the lack of positive selection due to wing overlap preventing selection by visual predators. The hypothesis behind this survey is nevertheless not entirely clear, since eyespot can also be involved in deimatic behaviour, promoting the evolution of eyespot in wing areas covered by the other wing. A more thorough analysis including comparisons of the eyespots shapes and size in covered vs. uncovered areas of both ventral and dorsal sides of the wings would probably bring more solid conclusions on the effect of wing overlap on the evolution of eyespot number in Nymphalid butterflies.

Line 242: the homology of FW and HW cells should be explained in the introduction, and assuming these common developmental constraints, explain that the working hypothesis for the difference between HW and FW distribution of eyespots is interpreted as a result of selection. This working hypothesis is not mentioned in the introduction.

Line 244: again replace the term 'asymmetry'

Line 230,247,291: The title of the result subsection must explicitly relate to the hypothesis tested or the results found rather than repeating the methods used. I would probably report the behavioural experiment looking at the target of selection (arena experiment) before the estimation of the impact of phenotype on fitness (microcosm experiment).

Line 265: following up with my sex-ratio survey suggested above, did butterfly females die sooner than male?

Line 269: Was the number of eggs computed per butterfly ALIVE? If so, the higher mortality on spotty should not be accounted for the reduced fertility?

Fig2C: the high attack rate on the green part on the leading edge of the FW of the wt individuals is not mentioned, did you have an interpretation for this surprising pattern? (see also my related comment on the material and method section).

Line 288-289: The results of this experiments are thus in line with the hypothesis of increased predation associated with increased number of eyespots, but do not show an increased number of damages on the FW in the spotty individuals, in line with the arena experiment.

Line 295: 'In eight OUT of the 69 attacks'

Line 296-300: This should be moved to the Material and Method section.

Line 309: This result thus contradicts the deflection hypothesis, the increased number of eyespots in the FW of spotty does not increase the attack toward the FW, this should be more explicitly

stated here.

Discussion:

Line 318: As explained below, this general sentence is useless and misleading and should be removed.

Line 321: replace 'asymmetry' by a more precise term.

Line 323: here phenotype with more eyespots in the FW were compared with phenotype with less eyespot in the FW but also less eyespots in total. It is thus not possible to actually conclude about the effect of uneven distribution. Moreover, in both experiments, the increased number of eyespots in the FW of spotty does not especially increase the attack toward the FW, but it does increase the attacks on both wings (arena experiments) and lead to an increase attack rates. This is in line with the hypothesis that increase number of eyespots might increase conspicuousness.

This result however contradicts the deflection hypothesis, because the spotty phenotype was not more heavily attacked on the FW. This should be properly discussed and the trade-off between conspicuousness and deflection in the evolution of eyespots would be explained and discussed.

Lines 350-351: This interpretation is not supported since only two eyespots distribution were tested, and is not related to the general question of the evolution of eyespot number and distribution on butterfly wings, this should be removed or rephrased.

Lines 352-363: The hypothesis that wing damages on the FW are probably more detrimental to flight (and therefore to survival and/or reproductive success) is supported but seems a bit off-topic since spotty butterflies were not necessarily more attacked on the FW.

Lines 365-376: As mentioned below, tracking down the sex-ratio throughout the microcosm experiment would help understanding what is driving the decrease in egg-laying in the cage with spotty butterflies. Once again, I am not familiar with *Bicyclus anynana* but in many butterfly species, females struggle to escapes male in a cage, and I am not sure they are really able to express their preferences in such an experimental set-up (no choice experiment with a constant harassment from males).

Decision letter (RSPB-2020-2840.R1)

21-Mar-2021

Dear Dr Chan:

Your manuscript has now been peer reviewed and the reviews have been assessed by an Associate Editor. The reviewers' comments (not including confidential comments to the Editor) and the comments from the Associate Editor are included at the end of this email for your reference. As you will see, the reviewers have raised some concerns with your manuscript and we would like to invite you to revise your manuscript to address them.

We discourage multiple rounds of revision so we urge you to make every effort to fully address all of the comments at this stage. If deemed necessary, your manuscript will be sent back to one or more of the original reviewers for assessment. If the original reviewers are not available we may invite new reviewers. Please note that we cannot guarantee eventual acceptance of your manuscript at this stage.

When submitting your revision please upload a file under "Response to Referees" in the "File Upload" section. This should document, point by point, how you have responded to the reviewers' and Editors' comments, and the adjustments you have made to the manuscript. We

require a copy of the manuscript with revisions made since the previous version marked as 'tracked changes' to be included in the 'response to referees' document.

Research ethics:

Use of animals and field studies:

It is a condition of publication that you make available the data and research materials supporting the results in the article (<https://royalsociety.org/journals/authors/author-guidelines/#data>). Datasets should be deposited in an appropriate publicly available repository and details of the associated accession number, link or DOI to the datasets must be included in the Data Accessibility section of the article (<https://royalsociety.org/journals/ethics-policies/data-sharing-mining/>). Reference(s) to datasets should also be included in the reference list of the article with DOIs (where available).

Online supplementary material will also carry the title and description provided during submission, so please ensure these are accurate and informative. Note that the Royal Society will not edit or typeset supplementary material and it will be hosted as provided. Please ensure that

the supplementary material includes the paper details (authors, title, journal name, article DOI). Your article DOI will be 10.1098/rspb.[paper ID in form xxxx.xxxx e.g. 10.1098/rspb.2016.0049].

Please submit a copy of your revised paper within three weeks. If we do not hear from you within this time your manuscript will be rejected. If you are unable to meet this deadline please let us know as soon as possible, as we may be able to grant a short extension.

Best wishes,
Dr Locke Rowe
Editor, Proceedings B
mailto:proceedingsb@royalsociety.org

Editor:

Comments to Author:

Three referees have read this version of your ms. One is happy with it, but the other two have raised some concerns that I believe are valid. The main concern in my opinion is that your results are over interpreted. There are a number of concerns. For example, even in the title it is stated that 'predation explains' across the group, when you do not have the data to support this. In short you have taken a large step toward rejecting the hypothesis that degree of overlap is correlated to spot differences between wings (though see below), but that does not by default lend support to the hypothesis that predation does account for it. The support for predation is limited to your experiments with one species. Likewise, the referee points out that the experiments do not directly provide a selective explanation for the evolution of an uneven number of eyespots in the ventral side of forewing vs. hindwing. Similarly, the use of photos to assess overlap has some weaknesses. All of these suggest that you should be much more critical of your own results, and much more circumspect in your conclusions (including in the title). In my view this approach will strengthen, not weaken the impact of your work. Referee 4 has several additional suggestions that should improve the clarity and impact of the paper.

Reviewer(s)' Comments to Author:

Referee: 1

Comments to the Author(s)

Congratulations for your interesting paper.

Referee: 3

Comments to the Author(s)

The author presents a very well written manuscript on the asymmetric distribution of eyespots in nymphalid butterfly wings. The work is embedded in relevant literature and the manuscript is nicely structured and well-organized. Their results of a comparative study and two experiments provide support for the hypothesis that selective predation plays a role in the evolution and maintenance of eyespot asymmetry (between fore- and hindwings). I have read the current version of the manuscript in detail and gone over the authors responses to reviewers comments. I believe that the authors have satisfactorily addressed all comments. I agree that the results of this study are interesting and important, and they deserve to be published.

Having said this, I think statements such as 'predation explains' in the title, and 'predation is the primary driver behind the asymmetric eyespot distribution observed in *B. anynana* and most other nymphalid species' in the Discussion are not justified by the data and should be tuned-down. The work indeed reveals that predation may play a role (and this is a valuable contribution in its own right), but more data is needed (e.g. confirmation from other species and other predators, etc) to validate that predation is in fact the 'primary driver'.

As a more general comment, I believe that the comparative study as conducted here, using few images from field guides, excluding intraspecific variation and phylogenetic effects, is far from ideal. Nymphalidae demonstrate a plethora of resting behaviours, and even within a single *Mycalesina* species there is ample variation in the relative exposure of the forewing (and even the display of dorsal vs ventral). *B. anynana*, for example, only 'folds away' their forewing after resting for a while (and this state will not often be presented in field guides because it is less informative from a taxonomical perspective). Nevertheless, I realize these data are very hard to come by, and I am confident that the observed pattern (i.e. wing overlap is a not a limiting factor) will hold.

Minor comments:

For the arena experiment, giant Asian mantis nymphs *Hierodula* sp. were trapped IN a park along Upper Serangoon Road, Singapore. THEY were reared to adulthood [in the laboratory] AND only female mantids were used.

\ Erik van Bergen

Referee: 4

Comments to the Author(s)

This manuscript aims at exploring the evolutionary forces involved in the uneven number of eyespots in the forewing vs. hindwing observed in many Nymphalid butterflies. First, the degree of wing overlap on the ventral forewing pattern of 275 species is explored and the authors conclude that the reduced number of eyespots on the ventral side of the forewing probably not rely on the lack of positive selection due to wing overlap preventing selection by visual predators.

The manuscript then also described two interesting experiments with mantid predators, comparing the wing areas attacked, as well as the fitness (survival and fecundity) of *B. anynana* butterflies differing in the number of forewing eyespots. Butterflies with increased number of eyespots (spotty mutants) suffers from increased attack rate and reduced fecundity, but the experiments exploring the mantid behaviour reveals that the increase number of eyespots in the forewing does not result in increased number of attacks in the forewing but rather in an increased attack number on both wings. These experiments thus do not directly provide a selective explanation for the evolution of uneven number of eyespots in the ventral side of forewing vs. hindwing. The main caveats being that the experiments rely on the comparison of two colour-patterns differing not only in the FW/HW distribution of eyespots but also in their total number of eyespots.

Nevertheless, the reported experiments bring new insights on the predation induced by eyespots located on the ventral side of butterflies, I would thus recommend bringing clarifications on the hypotheses tested and general conclusions brought by the experiments.

Specific comments:

Title and throughout the text: the term 'asymmetry' to refer to difference between the forewing and the hindwing is misleading. Asymmetry would rather be appropriate to compare right and left wings. I would recommend using more precise term such as 'uneven eyespots distribution between fore and hindwings'.

Abstract

Line 17: 'Eyespots are anti-predator defence in many butterflies': this general statement is misleading, although the eyespots have been shown to deflect attacks in some cases, the effect of eyespots on predator behaviour is not straightforward and remains to be investigated, and this study contributes to this understanding, I would avoid such over-general and vague sentence and rather start with the evolutionary question investigated in this study.

Line 17-18: It is also misleading to report the result of a previous study in the abstract.

Line 22: 275 nymphalid species ALL EXHIBITING EYESPOTS

Line 23: 'does not limit' is unclear. Did you mean that covered areas do display eyespots in many species? Here, and throughout the manuscript, it is a bit unclear why and how would the

coverage of the forewing by the hindwing would prevent the evolution of eyespots? Is it because, those partially covered eyespot are less likely to be seen by predators, and therefore less likely to be under positive selection? This needs to be more explicit in the abstract and in the introduction.

Line 24: The rationale behind the two experiments should be provided here and in the introduction as well, one experiment allows to compare the fitness associated with spotty vs. wild types phenotype, while the other experiment focuses on predator behaviour and the targeted regions of wings.

Line 28-30: This interpretation of the results is not entirely correct: butterflies with increased number of eyespots (spotty mutants) suffers from increased attack rate and reduced fecundity, but the experiments exploring the mantid behaviour reveals that the increase number of eyespots in the forewing does not result in increased number of attacks in the forewing but rather in an increased attack number on both wings. These experiments thus do not directly provide a selective explanation for the evolution of uneven number of eyespots in the ventral side of forewing vs. hindwing. The main caveats being that the experiments rely on the comparison of two colour-patterns differing not only in the FW/HW distribution of eyespots but also in their total number of eyespots. The experiments clearly show that individuals with more eyespots suffers from increase attacks, but does not allow to conclude about the selection of uneven number in FW/HW.

Introduction:

Line 40: the term 'function' sounds finalistic and is often misleading, because most trait evolve under multiple selective pressures. The evolution of eyespots can result from trade-offs between conspicuousness and deflection for instance, so that this sentence should be entirely rephrased to explain the selective pressures at play and highlight that their relative contribution remains to be studied.

Line 47: replace 'the eyespots' attack deflection function' by 'deflection of the attacks on the eyespots, located in less-vital parts of the body'.

Line 58: removed 'it is interesting that'.

You may also detail why you focus on the ventral side of the wings and do not investigate the effect coverage of wing on the evolution of eyespots on the dorsal side as well.

Line 63: replace 'be able to function as a continuously-visible signal' by 'be continuously-visible'

Line 64: the term 'useful' is definitely finalistic, and this misleading sentence prevent a proper description of the selective force involved: did you mean prevent positive selection due to attack deflection? The deimatic effect, although poorly studied should probably not be neglected, and is likely to depend on the relative size and shape of the different eyespots, that has not been investigated here. It is particularly surprising to discard this hypothesis so quickly, because as shown on fig.1A, the biggest and most conspicuous eyespot observed on the ventral side of the wings of *B. anynana* is located on the forewing, and in a partially covered area.

Line 67 and 73: As mentioned above, the term 'asymmetry' is not entirely appropriate here.

Line 92-95: The questions investigated in the two experiments should be more explicit: the microcosm experiments allow to estimate the effect of the two phenotypes on the fitness of the individuals by comparing survival and reproductive success with and without predators, while the arena experiments precisely investigates the region of the wings targeted by predator's attacks, and therefore the selection acting on the distribution of eyespots.

I would also advise to present first the behavioural experiment and then the impact on fitness.

Materials and Methods

Line 104-105: Is there a reason for ignoring behaviour variation within species? The mean position covered could be used? Would it change the results or is this variation observed for a large or a small proportion of the species? These data could be of interests for a large number of entomologists, the authors indeed provide a Dryad number, but I was not able to access it and therefore could not check them (maybe because the data will be online only after the publication?).

Line 123-125: Is there any reason to use different predators in the two experiments? And to use female only in the arena experiments while using both sex in the microcosm. As mentioned in the introduction, both are probably relevant predators, but it makes the two experiments less related to each other.

Line 126-127: Did the wild-type and mutant share the same genetic background? Are spotty butterflies related to each other? An increase in relatedness among individuals could affect their fertility within a cage.

Line 143-144: Was wing damage quantified in both control and experimental cages? Did the butterflies have wing damage in a cage without predators (because they rub their wings against the cage walls for instance). I guess Fig. 2C reports damage in cages with predators? It would be important to disentangle the effect of the damage caused by predators from the damage due to aging and rubbing against the cage walls.

Line 147-148: The rationale behind the use of the 0.5 scoring is not entirely clear. Here, it is hard to assess whether a mark spreading two wing cells was due to a single attack or a succession of several attacks and subsequent wing rubbing. It would probably be more conservative to have a simpler binomial score for each wing cell describing whether it was intact or damaged. For a quantitative measure, you would need to compute, for each cell, the proportion of the wing cell surface removed.

Line 166: Was the number of eggs laid divided by the number of butterflies alive during each day? Wouldn't it be more relevant to control for the number of females still alive, because only females are able to lay eggs? I am not familiar with *B. anynana*, but for other butterfly species, females do not remain virgin in a cage with males, and the number of eggs will thus strongly depend on the sex-ratio, females being frequently harassed by males. It would be relevant to control for the changes in sex-ratio throughout the experiments when studying the number of eggs produced.

Line 167: 'problems with error distribution', this must be more precise.

Line 171: the percentages reported on the heat map (fig. 2C) are not straightforward: for example, the green part on the leading edge of the FW: does it mean that twice as much wt individuals were damaged at this location with respect to spotty individuals? Is it standardized by the total number of individuals damaged at this location? Please provide more detail on the way these percentages were computed.

Line 188 and line 131: the general purpose of each of the two experiments should be given at the beginning of the sections, for the reader to immediately know what will be tested in each experiment.

Line 216: I did not understand why you studied the first four strikes while each mantid was tested 8 times (with four spotty and four wt butterflies, as mentioned on line 208).

Line 217: Does the 'type of strike' relate to the position targeted by the mantid (as described on line 296-300)? If so, this should be clearly stated here.

Line 219: Are 'first strike categories' similar to the 'type of first strike'? It would be easier to define those types or categories and use a consistent vocabulary.

Results

Line 235: 'Wing overlap is a limiting factor': this is an interpretation, Does it mean that in 12.8% of species, no eyespots were observed in the FW zone covered by the HW? This should be precisely described. Similarly, on line 238, it is rather difficult to understand what is the result for the 240 remaining species? Do they display eyespots in the FW zone covered by the HW?

If I understood correctly, only a limited number of species had no eyespots in the covered part of the forewing despite displaying eyespots on the rest of the wings. The authors conclude that the reduced number of eyespots on the ventral side of the forewing probably does not rely on the lack of positive selection due to wing overlap preventing selection by visual predators. The hypothesis behind this survey is nevertheless not entirely clear, since eyespots can also be involved in deimatic behaviour, promoting the evolution of eyespots in wing areas covered by the other wing. A more thorough analysis including comparisons of the eyespot shapes and size in covered vs. uncovered areas of both ventral and dorsal sides of the wings would probably bring more solid conclusions on the effect of wing overlap on the evolution of eyespot number in Nymphalid butterflies.

Line 242: the homology of FW and HW cells should be explained in the introduction, and assuming these common developmental constraints, explain that the working hypothesis for the difference between HW and FW distribution of eyespots is interpreted as a result of selection. This working hypothesis is not mentioned in the introduction.

Line 244: again replace the term 'asymmetry'

Line 230,247,291: The title of the result subsection must explicitly relate to the hypothesis tested or the results found rather than repeating the methods used. I would probably report the behavioural experiment looking at the target of selection (arena experiment) before the estimation of the impact of phenotype on fitness (microcosm experiment).

Line 265: following up with my sex-ratio survey suggested above, did butterfly females died sooner than male ?

Line 269: Was he number of eggs computed per butterfly ALIVE ? If so, the higher mortality on spotty should not be accounted for the reduced fertility ?

Fig2C: the high attack rate on the green part on the leading edge of the FW of the wt individuals is not mentioned, did you have an interpretation for this surprising pattern ? (see also my related comment on the material and method section).

Line 288-289: The results of this experiments are thus in line with the hypothesis of increased predation associated with increased number of eyespots, but do not show an increased number of damages on the FW in the spotty individuals, in line with the arena experiment.

Line 295: 'In eight OUT of the 69 attacks'

Line 296-300: This should be moved to the Material and Method section.

Line 309: This result thus contradicts the deflection hypothesis, the increased number of eyespots in the FW of spotty does not increase the attack toward the FW, this should be more explicitly stated here.

Discussion:

Line 318: As explained below, this general sentence is useless and misleading and should be removed.

Line 321: replace 'asymmetry' by a more precise term.

Line 323: here phenotype with more eyespots in the FW were compared with phenotype with less eyespot in the FW but also less eyespots in total. It is thus not possible to actually conclude about the effect of uneven distribution. Moreover, in both experiments, the increased number of eyespots in the FW of spotty does not especially increase the attack toward the FW, but it does increase the attacks on both wings (arena experiments) and lead to an increase attack rates. This is in line with the hypothesis that increase number of eyespots might increase conspicuousness. This result however contradicts the deflection hypothesis, because the spotty phenotype was not more heavily attacked o the FW. This should be properly discussed and the trade-off between conspicuousness and deflection in the evolution of eyespots would be explained and discussed.

Lines 350-351: This interpretation is not supported since only two eyespots distribution were tested, and is not related to the general question of the evolution of eyespot number and distribution on butterfly wings, this should be removed or rephrased.

Lines 352-363: The hypothesis that wing damages on the FW are probably more detrimental to flight (and therefore to survival and/or reproductive success) is supported but seems a bit off-topic since spotty butterflies were not necessarily more attacked on the FW.

Lines 365-376: As mentioned below, tracking down the sex-ratio throughout the microcosm experiment would help understanding what is driving the decrease in egg-laying in the cage with spotty butterflies. Once again, I am not familiar with *Bicyclus anynana* but in many butterfly species, females struggle to escapes male in a cage, and I am not sure they are really able to express their preferences in such an experimental set-up (no choice experiment with a constant harassment from males).

Author's Response to Decision Letter for (RSPB-2020-2840.R1)

See Appendix B.

Decision letter (RSPB-2020-2840.R2)

04-May-2021

Dear Dr Chan

I am pleased to inform you that your manuscript RSPB-2020-2840.R2 entitled "Predation favours *Bicyclus anynana* butterflies with fewer forewing eyespots" has been accepted for publication in Proceedings B.

The Associate Editor has recommended publication, but also suggests some minor revisions to your manuscript. Therefore, I invite you to respond to these comments and revise your manuscript. Because the schedule for publication is very tight, it is a condition of publication that you submit the revised version of your manuscript within 7 days. If you do not think you will be able to meet this date please let us know.

Sincerely,
Dr Locke Rowe
Editor, Proceedings B
<mailto:proceedingsb@royalsociety.org>

Associate Editor:
Board Member
Comments to Author:

I would like to thank the authors again for making such a good effort at attending to each of the comments raised by the reviewers. I think that you have done well in the text by clarifying issues relating to the attack rates on the forewings in the arena experiment and also generally being more cautious about your interpretation of the data.

I have a few very minor comments to make below:

L30: Sounds almost as though one of the findings was that "carrying more forewing eyespots is detrimental because forewings are more important for flight." This is not really something that the ms investigated. I suggest making this clearer: "These results suggest that forewing eyespot number in *B. anynana* is limited by predation pressure. This may occur if attacks on forewing eyespots have more detrimental consequences for flight than attacks on hindwing eyespots."

L65: Deimatic is a great word but I am not sure that most people would be familiar with its meaning. I would consider this jargon.

L66: Not 100% clear why: Perhaps try "Second, higher numbers of eyespots on the hindwings may be advantageous if they deflect predator attacks away from the forewings which are more important for flight."

L85: Are spots likely to make the wing more conspicuous or do they dupe the predator into thinking they are striking at an important body part?

L94: delete "for"

L133: not sure what you mean by "subject to similar numbers of male and female mantids being assigned to each form"

L140: delete "respectively"

L250: not sure if "censored" is the word you are looking for

L275: in control cages there was no difference in the wing damage...

L281: Since most people use 0.05 as their cut-off for significance (arbitrary as it is) most people would consider this as being marginally insignificant not marginally significant. Perhaps try: "These differences border on significance for the forewings ($P = 0.08...$)"

L300: again, this is not marginally significant, but bordering on significance.

L302-308: This is all one sentence and is almost impossible to follow. Break it up.

Author's Response to Decision Letter for (RSPB-2020-2840.R2)

See Appendix C.

Decision letter (RSPB-2020-2840.R3)

06-May-2021

Dear Dr Chan

I am pleased to inform you that your manuscript entitled "Predation favours *Bicyclus anynana* butterflies with fewer forewing eyespots" has been accepted for publication in Proceedings B.

Data Accessibility section

Open Access

Paper charges

Sincerely,

Proceedings B

Appendix A

Evolutionary Development Laboratory,
Department of Biological Sciences,
National University of Singapore,
14 Science Drive 4, Blk S2 Level 1,
Singapore 117543

email: ianchan@nus.edu.sg

19 Jan 2021

Dear Prof. Barrett and Prof. Rowe,

Through the electronic submission system, I have submitted our revision to the Research Article “Predation explains asymmetric eyespot distribution across wings of nymphalid butterflies” by Ian Z.W. Chan, Zhe Ching Ngan, Lin Naing, Yueying Lee, V Gowri and Antónia Monteiro (MS Reference Number: RSPB-2020-2840). We are grateful for this opportunity to revise our ms, and would like to thank you and the reviewers for the constructive comments provided in your email of 22 Dec 2020.

In this document, we provide our responses to the issues raised. Text in black are Editor’s or Referee’s comments and text in blue are our responses. Unless otherwise stated, all line numbers refer to the clean revised ms (i.e. where all tracked changes have been accepted).

Briefly, our main responses and changes are:

- 1) We now describe our experimental procedure in more detail, as requested by Referee 1;
- 2) We now discuss various issues and have consulted with a statistician on accounting for repeated measurements by the mantids in our arena study, as suggested by Referee 2;
- 3) We have uploaded the datasets and R code previously in the Supplementary Material onto Dryad, as required by the journal; and
- 4) We made various parts of the ms more concise to meet the space constraints.

Thank you again for this opportunity to submit our revised ms. We hope that you find our responses satisfactory and that the ms is now acceptable for publication in your journal *Proceedings B*.

Yours sincerely,

Dr. Ian Chan
Research Fellow

Response to Editor

This is the second time I am handling this manuscript. The first time, I read the manuscript thoroughly and suggested multiple additions and changes before the manuscript was sent for review. These changes were made by the authors and the manuscript was sent to two reviewers. I now report back on the findings of the two reviewers, both of whom thought that the manuscript was interesting. Despite the general interest by both reviewers, both also have some critical comments for the authors to address. The first reviewer outlines various places in the methods where crucial information is missing, making it impossible for the experiments to be accurately replicated. I implore the authors to address these comments carefully and include the relevant information.

We now describe our methodology in greater detail, as requested by Referee 1. We write about this in greater detail in our response to the reviewer below.

Among other points, the second reviewer writes that it is possible that spot number is not the only difference between the two butterfly forms and so it is possible that the observed results are not due to differences in spot number. Here I suggest that some of the points brought up by reviewer 2 are critically discussed in the discussion.

As suggested, we have now added text to discuss the points raised by Referee 2. This is described in detail in our response to the reviewer below.

This reviewer also wrote to me that “it would be good for a specialist statistical reviewer to check whether the authors have appropriately taken into account multiple measurements from individual mantids in the ‘Arena’ experiment (the whole experiment only included 10 mantids in total).” I realize that the authors have used mantid ID as a random factor in their models, but a response to this point will also help me decide on how to proceed with the manuscript.

There is support in the literature for the use of a random effect in a binomial generalised linear mixed model to account for repeated measurements using the same participant/subject. Quené & Van den Bergh (2008) analysed a study where 36 human participants repeatedly identified sounds (producing a binary “hit-or-miss” response variable, similar to our pairwise comparison between first-strike locations) and showed that participant ID, as a random effect, was able to account for variation caused by repeated measurements using the same participants (analogous to our usage of “Mantid ID” to account for repeated trials using the same mantid).

We have also consulted a statistics lecturer at our department, Prof. L. Roman Carrasco, on this issue. After reviewing our study, Prof. Carrasco’s considered opinion is that the analyses (i.e. multinomial and binomial generalised mixed models) are appropriate for the arena experimental data, and that the inclusion of “Mantid ID” as a random effect is sufficient to account for the different “personalities” of the 10 mantids, including, for example, differences in how each mantid learns.

References

Quené, H., & Van den Bergh, H. (2008). Examples of mixed-effects modeling with crossed random effects and with binomial data. *Journal of Memory and Language*, 59(4), 413-425.

Please respond carefully to all of the reviewer comments.

Thank you for your time in facilitating this review. Please find below our responses to the issues raised by the two reviewers.

Response to Referee 1

This is a very interesting paper. The main aim was to contrast two hypotheses to explain the evolutionary maintenance of asymmetric eyespot distribution between fore- and hindwings. One hypothesis involves constraints and the other natural selection derived from visually oriented predators. The results from a comparative study and two experiments support the last hypothesis.

Thank you. We are very encouraged by your interest in our study.

My main concerns with this manuscript are related to the Methods. In particular, more information is needed to understand how the comparative study and the arena experiment were done and, thus, to judge its results and interpretation. More general, I consider that these two studies cannot be replicated with the information provided in the manuscript.

First, with respect to the study comparing the part of the forewings that is covered by the hindwings (when perching) using photographs taken from different public repositories, I wonder how is it possible to determine what this area is for the whole species and why these estimates are presented as error free. I have experience observing perching butterflies in the field and recently, for reasons that have nothing to do with the subject of this manuscript, with one of my colleagues, I have been looking at photographs of different species of butterflies (not nymphalids, I must admit) that perch (and feed and oviposit) with their wings closed and I can attest that the front wing area covered by the hindwings varies within species (frequently with time in the same individual), and sometimes varies a lot. I think that it is necessary to explain clearly how these data were obtained, how much intraspecific variation was observed, the number of photographs on which the estimates are based, etc., to allow the reader to judge the results presented.

We agree with the reviewer that forewing overlap can vary within a species and even within an individual over time. To mitigate the effects of this variation, we applied the following guidelines in our analysis. First, we surveyed as many images as possible per species (ranging from one to ten, but typically two to four) and chose the most common degree of overlap. Second, in cases where there was a tie, we chose the more conservative observation (i.e. where more sectors are covered). It is likely that the data will change as more images become available in future but we believe that we provide here a conservative estimate of the number of forewing sectors exposed at rest in these 431 species. We include the URLs of the images on which we based our observations in the data file uploaded to the Dryad repository, and have now added the information above to the M&Ms.

Added Text

Lines 102-107 (M&Ms): ‘Because there can be variation in the degree of overlap within species and individuals, we applied the following guidelines in our analysis. First, we surveyed as many images as possible per species and chose the most commonly observed degree of overlap. Second, in cases where there was a tie, we chose the more conservative observation, i.e. the one where more sectors are covered. Hence, the estimate we provide here for the number of exposed forewing sectors at rest is likely to be a conservative one.’

Second, in the arena experiment it is mentioned that encounters between one mantis and one butterfly were staged and that a total of 10 mantids were used, each was exposed in a random sequence to 8 butterflies (4 wild type and 4 spotty). To replicate this experiment it is needed to explain how was this done. For example, one mantis was exposed to the complete sequence of butterflies the same day or to how many per day? Different mantids were exposed to butterflies at the same time or there was only one arena and the different mantises were used in alternate moments? etc.

We constructed only one arena, so only one trial could be conducted at a time. After an initial two-day starvation, we conducted one trial each for all 10 mantids. We then starved all the mantids for another two-day period before carrying out one trial each for all 10 mantids again. This was repeated until eight trials per mantid had been completed. As all mantids performed their trials over the same time frame, we expect long-term effects (such as aging) to be consistent across the 10 mantids. We also account for differences in mantid personality in the statistical analysis using a random effect in our models. We have now added these details to the text.

Edited Text	Original Text
Lines 205-208: ‘A total of 10 mantids were used, and one round of trials comprised 10 trials: one per mantid in random sequence. After the first round, the mantids were again starved for two days before conducting the second round of trials. This process was repeated until eight rounds were completed.’	‘A total of 10 mantids were used in the experiments.’

Finally, in lines 131-132 it is mentioned that only female mantids were used in the experiments, although this is clearly not the case in the “microcosm experiment” (mantid sex was used as an effect in the analyses). In the “arena experiment” the sex of the mantids is not mentioned and it is not included in the analyses. It is possible that it was in this experiment that only females were used.

The reviewer is correct in their interpretation: both male and female mantids were used in the microcosm experiment, whereas only female mantids were used in the arena experiment. We apologise for the confusion and have now amended the text to make this clearer.

Edited Text	Original Text
Lines 121-126: ‘For the microcosm experiment, male and female Polyspilota mantids were purchased from BugzUK (34 North Walsham Road, Norwich, UK) at the 6 th instar and reared until adulthood for use. For the arena experiment, giant Asian mantis nymphs Hierodula sp. were trapped from a park along Upper Serangoon Road, Singapore and reared to adulthood when only female mantids were used. Their diet (fruit flies,	‘ Polyspilota mantids (for the microcosm experiment) were purchased from BugzUK (34 North Walsham Road, Norwich, UK) at the 6th instar and reared until adulthood for use in experiments. Giant Asian mantis nymphs Hierodula sp. (for the arena experiment) were trapped from a park along Upper Serangoon Road, Singapore and reared individually to adulthood. Their diet (fruit flies, grasshoppers and mealworms) was

grasshoppers and mealworms) was chosen to avoid exposure to butterflies or eyespots prior to experiments.’	carefully chosen to not expose them to butterflies or eyespots prior to the experiments. Only adult female mantids were used.’
--	--

Regarding the Results section:

In the results relative to the microcosm experiment, butterflies with different number of wing spots are referred to as “forms”. I think this can be confusing, as for example when it is said that “in control cages, butterfly form did not affect survival”.

In the excerpt quoted by the reviewer, we do mean to compare the survival of wildtype versus Spotty butterflies in control cages, as specified in the second half of the same sentence: ‘there was no significant difference in the longevity of Wt (29.4 days) and Spotty (27.2 days) individuals ($P = 0.48$)’ (lines 256-257). Throughout the ms, we refer to the wildtype and Spotty phenotypes, with different number of wing spots, as two different forms. This is also consistent in the R code used for the data analysis.

In lines 274-277 the authors suggest that the difference in “fecundity” observed between Wt and Spotty butterflies is “likely driven by the shorter lifespan” of the Spotty butterflies. However, the data they present show differences in the number of eggs laid per day and I think that Figure 2b (lower panel) also does not support their interpretation. The reduced egg-laying rate might be a product of more mantis “harassment” on Spotty butterflies.

We agree with the reviewer on this point and have now amended the text.

Edited Text	Original Text
Lines 268-270: ‘This lower fecundity is likely driven by increased predation pressures on Spotty relative to Wt animals.’	‘This lower fecundity is likely driven by the shorter life span of Spotty animals relative to Wt animals.’

Finally, I wonder what proportion of attacks in the arena experiment resulted in predation and if there was a difference in this proportion between Wt and Spotty butterflies.

Of the 74 trials completed, 69 attacks were recorded (35 on Wt and 34 on Spotty) and five trials resulted in no attack (four on Wt and one on Spotty). This would appear to agree with our microcosm experiment results in suggesting that Spotty is more likely to be attacked, however more observations would be needed for us to be confident with this claim. Of the 69 attacks recorded, eight successful escapes were observed, four by Wt and four by Spotty. We have now included these data in the ms.

Edited Text	Original Text
Lines 292-296: ‘Eight mantids successfully completed all eight trials. Two mantids died prematurely, completing four trials (two Wt,	‘Most mantids successfully completed all eight trials assigned to them. Two mantids died prematurely, completing only four trials

two Spotty) and six trials (four Wt, two Spotty). A total of 69 attacks were recorded (35 on Wt and 34 on Spotty) and five trials (four Wt, one Spotty) resulted in no attack. In eight of the 69 attacks observed (four Wt, four Spotty), the butterfly subsequently managed to escape.’

(two with Wt and two with Spotty butterflies) and six trials (three with Wt and three with Spotty butterflies) respectively. A total of 69 attacks were recorded (35 on Wt and 34 on Spotty butterflies) and five trials resulted in no attack.’

With respect to the Discussion:

In lines 363-365 I found two confusing assertions:

(1) “[M]antids predominantly shifted from targeting the hind wing only in Wt to targeting both wings in Spotty” – This assertion gives the impression that mantids attack predominantly the hindwings of Wt butterflies, which is not the case.

(2) “As the hindwing is targeted in both the forms, there would be a bigger difference on the forewings and a smaller difference (if any) on the hindwings” – I found this confusing because mantids barely directed attacks to the hind wing in Spotty butterflies (figure 3c), although they directed the larger proportion of their attacks to the eyespots of both wings. Also, Wt butterflies were most attacked in the eyespots of the forewing area. I feel that the confusion will disappear if the drafting of these assertions is improved.

We chose to focus on discussing the differences between wildtype and Spotty (as opposed to which parts were attacked most frequently *per se*). The principal difference (illustrated in figure 1 in this response document) is that the “forewing eyespots” and “hindwing eyespots” categories become smaller whilst the “eyespots on both wings” category becomes bigger in Spotty. This suggests that the main change in how mantids attacked Spotty (versus wildtype) was a decrease in attacks on the forewings alone and on the hindwings alone, and a corresponding shift towards attacking both wings simultaneously.

Figure 1. Arena experiment results (extracted from Figure 3c in the ms). Comparing Spotty to Wt, the results suggest that the mantids switched away from striking the “forewing eyespots” alone and “hindwing eyespots” alone in Wt (these two categories, especially the hindwing, are much reduced in Spotty) to attacking the “eyespots on both wings” in Spotty (this category more than doubled to almost 40% in Spotty). The proportion of first strikes on the “body” in both forms remains similar.

The decrease in attacks on “hindwing eyespots” in Spotty (2.9% in Spotty versus 17.1% in wildtype), however, was more pronounced than the decrease in attacks on “forewing eyespots” (29.4% in Spotty versus 40.0% in wildtype). This may be explained by an innate preference in this species of mantid to target the forewings (as evidenced by attacks on the forewing being the most common category in wildtype, which we now discuss in the ms), possibly because the largest of all the ventral eyespots is found on the forewing. Hence, the comparison between “hindwing eyespots” and “eyesspots on both wings” produced the only clearly significant difference between wildtype and Spotty in the statistical analysis of these arena experiments.

We have now edited the text to make this line of thought clearer.

Edited Text	Original Text
Lines 340-348: ‘The principal differences in first strike locations on Spotty relative to Wt butterflies are: (i) a decrease in attacks on the forewing alone and on the hindwing alone, and (ii) a corresponding shift to attacking both wings simultaneously in Spotty (figure 3c). The decrease, however, is more pronounced on the hindwing. This may be explained by a naïve preference for targeting forewings, as evidenced by attacks on the forewing being the most common in Wt, perhaps because this is the location of the largest ventral eyespot. Hence, the comparison between “hindwing eyespots” and “eyesspots on both wings” produced the only clearly significant difference between Wt and Spotty in the statistical analysis of these arena experiments.’	‘The only significant difference in the multinomial regression was between the proportions of attacks on hindwings versus on both wings, suggesting that mantids predominantly shifted from targeting the hindwing only in Wt to targeting both wings in Spotty (figure 3c). As the hindwing is targeted in both the forms, there would be a bigger difference on the forewings and a smaller difference (if any) on the hindwings.’

In conclusion, although I have doubts about the comparative study, I think that the results of the two experiments support the hypothesis that selective pressures derived from predators are at least partially responsible for the evolutionary maintenance of asymmetric eyespot distribution between fore- and hindwings and deserve to be published, although more details of the experimental procedures need to be provided.

Thank you. We have now provided additional detail on our experimental procedure.

Response to Referee 2

This is an interesting study. It was a pleasure to read the manuscript, and I think the analyses are well done. However, I have a couple of concerns regarding the conclusions drawn.

Thank you for this encouraging comment.

This sentence perhaps encapsulates what the authors infer as the main conclusion of the study: "Our data indicate that the location of eyespots on butterfly wings is important and suggest that predation is the primary driver behind the asymmetric eyespot distribution observed in *B. anynana* and most other nymphalid species.". The authors aim to explain why spots are more frequent on the hindwings compared to the forewings.

The study has two components. The first tests whether forewings have fewer eyespots than do hindwings simply because part of the forewing is hidden under the hindwing, and, therefore, there is an asymmetry in the area available to display to predators. Their analysis of spot numbers on the fore- and hindwings of butterfly species indicate that the asymmetry in eyespot numbers between the forewings and hindwings is not explained by space constraints.

The second component includes laboratory predation trials. Here, individuals with a mutant phenotype 'spotty' were pitted against wildtype butterflies. Compared to wild type ones, spotty butterflies had 2 additional eyespots on the forewing. The authors explicitly assume that there are no other differences between the two morphs apart from the 2 eyespots (L 83). Spotty butterflies had lower fitness as measured by multiple fitness proxies (e.g. how long it took the predator to consume the butterfly, fecundity, longevity, etc.).

My first concern here is about how well controlled (or not) the experiments were. Is the presence of the two eyespots the ONLY difference between the two morphs? a) Looking at Fig 1, spotty individuals seem to have eyespots with larger white centres. b) Furthermore, the eyespots in Cu1 and M3 appear to be fused together, although such a fusion of adjacent spots does not seem to be the case in the hindwing. I did an online search for Spotty and found a couple of other papers (in which the last author of the current study was also involved in) with images of this mutant. It appears that the fusion of adjacent eyespots is found in other spotty individuals. Of course, I cannot and should not draw any conclusions based on a few images. But, we cannot also exclude the possibility that spotty and wildtype butterflies differ in aspects other than $n/n+2$ eyespots. For e.g., the authors themselves discuss (L 51) *Parage aegeria*, a butterfly with eyespot number polymorphism, where there are behavioral differences between the morphs. The fusion of adjacent eyespots could create a new shape (as opposed to a circle) that may have fitness consequences. Therefore, I am not convinced that the differences between spotty and wild type can be attributed confidently only to the effect of the two eyespots.

There is indeed some degree of fusion between the M3 and Cu1 eyespots on the forewing of Spotty butterflies. In our lab populations, the outer gold rings are always fused whereas the inner black rings are fused in less than 10% of butterflies (compare the two wings in figure 2 in this document). Whilst the fusion of the rings does create a new shape, we argue that the unfused

black rings and the four white centres would still draw attacks to themselves in a manner similar to isolated eyespots (Chan et al. 2019). While eyespot shape, and eyespot fusions, would need to be examined in a future study, we believe that the additional eyespots in Spotty represent the type of eyespot number variation that we find (sometimes) in natural species of nymphalid.

Figure 2. Spotty wings from individuals in our lab population.

Left: the most common appearance in our lab population, where the gold rings of the M3 and Cu1 eyespots are fused but not the black.

Right: in less than 10% of butterflies, the black rings are also fused.

However, we agree with the reviewer that the fusion is more pronounced on Spotty forewings than on hindwings and that this, together with potential behavioural differences (which we did not notice but also did not specifically measure), could also have contributed to our observations. We have now discussed these two points and call for further research which account for these potential sources of variation.

Added Text

Lines 378-392: ‘Finally, it is possible that other factors could explain our results. Systematic differences in escape behaviour between Spotty and Wt individuals, which we did not notice during our experiments but did not specifically measure, could result in different survival rates, although previous work by Prudic et al. [7] indicated that the effects of eyespots on survival were independent of behaviour in wet and dry season forms of *B. anynana*. In addition, the M3 and Cu1 eyespots on the Spotty forewing are partially joined (the outer gold ring is always fused, and the inner black ring is fused in less than 10% of specimens), producing a new shape on the forewing which could affect mantid behaviour. This fusion, however, does not affect the conspicuity of the white eyespot centres. They should therefore still draw attacks to themselves in a manner similar to isolated eyespots [12]. Nevertheless, future research into this question should carefully control the appearance of eyespots to ensure that forms differ only in eyespot number, and not shape, e.g. by painting different numbers of identically-shaped eyespots, or gluing bits of wings with different numbers of eyespots (as in Prudic et al. [7]), onto the same butterfly genotype to produce two forms, one with more eyespots on the forewing and one with more eyespots on the hindwing.’

References

Chan IZW, Rafi FZ, Monteiro A. 2019. Interacting effects of eyespot number and ultraviolet reflectivity on predation risk in *Bicyclus anynana* (Lepidoptera: Nymphalidae). *J Insect Sci.* 19(6), 19. (doi: 10.1093/jisesa/iez123)

Furthermore, even if the difference in survival between the two morphs is explained solely by the additional two eyespots, I am not convinced that these results by themselves explain the broader pattern of forewing-hindwing asymmetry in butterflies. There could be a threshold effect in this particular species, wherein two additional eyespots are too many. In fact, this argument is bolstered by this statement in the Introduction "Furthermore, paper models of *B. anynana* with a greater number of small ventral marginal eyespots are more readily attacked, presumably because they are more conspicuous and attract more predator attention [6,12]". So, it is already known from previous work that too many eyespots on the ventral surface is not good (although having a few is better than having none).

The studies using paper models in the field are slightly counterintuitive to interpret because paper models do not exhibit escape behaviour after being attacked. Hence, while they are good tests of conspicuousness and attack risk (e.g. Ho et al. 2016), they do not give actual fitness data. Amongst the research we cite, studies using live prey show that prey items with more conspicuous eyespots were attacked more but also escaped more, survived longer and reproduced more (Prudic et al. 2015). We acknowledge that this distinction is not well explained in our Introduction and have now edited the text to make it clearer.

Edited Text	Original Text
Lines 52-56: ‘Furthermore, butterfly paper models with more small ventral marginal eyespots are more readily attacked, presumably because they attract more predator attention [6,12]. This increase in conspicuousness, however, can confer fitness benefits because the eyespots draw attacks to less important body parts, as demonstrated in studies using live prey items [7].’	‘Furthermore, paper models of B. anynana with a greater number of small ventral marginal eyespots are more readily attacked, presumably because they are more conspicuous and attract more predator attention [6,12].’

We also agree with the reviewer that extrapolating our conclusions to the whole of the nymphalids would be more strongly supported if these results can be reproduced in other species and now mention this in our Discussion.

Added Text
Lines 392-394: ‘The extrapolation of our conclusions to the Nymphalidae as a whole would be more strongly supported if future studies produce similar results in other species.’

References

- Ho S, Schachat SR, Piel WH, Monteiro A. 2016. Attack risk for butterflies changes with eyespot number and size. *Roy. Soc. Open Sci.* 3, 150614. (doi: 10.1098/rsos.150614)
- Prudic KL, Stoehr AM, Wasik BW, Monteiro A. 2015. Eyespots deflect predator attack increasing fitness and promoting the evolution of phenotypic plasticity. *Proc. R. Soc. B* 282(1798), 20141531. (doi: 10.1098/rspb.2014.1531)

To explain why eyespots are relatively infrequent on the forewing, a more appropriate experiment would be one involving a comparison between two treatments: 1) more eyespots on the forewing than on the hindwing 2) more eyespots on the hindwing than on the forewing. If the eyespot morphology is properly controlled, and fitness of individuals in the first treatment is higher, we would have stronger support for their hypothesis.

This is an exciting suggestion for a future experiment. We could apply paint to individuals of the same genotype (in a controlled and consistent manner, whilst also accounting for the smell of the paint) so that the two forms have exactly opposite eyespot distribution patterns. For example, starting from wildtype *B. anynana*, we could paint the butterflies so that one form has two eyespots on the forewing (M1 and Cu1) and four on the hindwing (M1, M2, M3 and Cu1) while the second form has four eyespots on the forewing (M1, M2, M3 and Cu1) and two on the hindwing (M1 and Cu1). Thank you for this idea. We now discuss it in the ms as mentioned in our edits above (lines 378-392).

Furthermore, deflective eyespots have been shown to function by attracting attacks towards the wing margin, such that a butterfly can escape with a torn wing. Forewings are more important than hindwings for flight and mating success (see L404). Thus, a torn forewing could be more detrimental than a torn hindwing in terms of flight performance and mating success. Thus, the asymmetry in eyespot frequency between forewings and hindwings may be explained by the indirect (through wing damage) effects of BOTH predation AND mate choice. This is something that the authors have not tested here, but should probably discuss in the Discussion.

We did not test for mate choice *per se* but agree that wing damage could also affect sexual signalling and discuss this issue in lines 365 – 376.

Some other comments

L39: Eyespots, bullseye patterns consisting of concentric rings of contrasting colours, are a common protective pattern in many lepidopteran species [3,4].

Please rephrase this. Eyespots are not always composed of concentric rings. In fact, many species in which eyespots have been studied -*Inachis io*, *Junonia* spp, etc - do not have eyespots with well defined concentric rings.

We have now rephrased this sentence according to the definition of an eyespot provided in Monteiro (2008).

Edited Text	Original Text
Lines 36-38: ‘Eyespots, “roughly circular patterns with at least two concentric rings or with a single color disc and a central pupil”	‘Eyespots, bullseye patterns consisting of concentric rings of contrasting colours, are a common protective pattern in many

[2], are a common protective pattern on the wings of many lepidopterans [3,4].’

lepidopteran species [3,4].’

References

Monteiro, A. (2008). Alternative models for the evolution of eyespots and of serial homology on lepidopteran wings. *Bioessays*, 30(4), 358-366.

L115 Second, we compared the number of exposed forewing versus hindwing eyespots in the species where the number of exposed forewing sectors was not a limiting factor. We defined this number as a limiting factor if it is less than the number of eyespots on the hindwing, i.e. even if all exposed forewing sectors bore an eyespot, there would still be fewer exposed eyespots on the forewing than the hindwing.

This was one of the few places where I had to read more than once, in order to try and understand. This was hard to follow - please consider rephrasing this.

This sentence has now been edited to make it clearer. We have also rephrased the corresponding results to make it simpler to understand.

Edited Text	Original Text
Lines 111-114: ‘Second, in species which have more exposed forewing sectors than hindwing eyespots (and which are therefore not limited to having fewer exposed forewing than hindwing eyespots), we compared the number of exposed forewing versus hindwing eyespots.’	‘Second, we compared the number of exposed forewing versus hindwing eyespots in the species where the number of exposed forewing sectors was not a limiting factor. We defined this number as a limiting factor if it is less than the number of eyespots on the hindwing, i.e. even if all exposed forewing sectors bore an eyespot, there would still be fewer exposed eyespots on the forewing than the hindwing.’
Lines 235-237: ‘Second, wing overlap is a limiting factor for the number of forewing eyespots in only 35 species (12.8%). These species have fewer exposed forewing sectors than hindwing eyespots.’	‘Second, within the same 275 species, the number of exposed forewing sectors is a limiting factor in only 35 species (12.8%).’

L 274. "Spotty butterflies laid half as many as eggs as Wt butterflies over the course of their lives (6.8 versus 276 13.4 eggs per butterfly per day; P = 0.022). This lower fecundity is likely driven by the shorter life span of Spotty animals relative to Wt animals."

This was perplexing to me. You measured the number of eggs laid per day. I presume you only counted the number of days a butterfly lived. If so, why should longevity affect the number of eggs laid per day? In fact, some animals live shorter but have higher per-day fecundity.

We agree with the reviewer on this point and have now amended the text.

Edited Text	Original Text
Lines 268-270: ‘This lower fecundity is likely driven by increased predation pressures on Spotty relative to Wt individuals.’	‘This lower fecundity is likely driven by the shorter life span of Spotty animals relative to Wt animals.’

Appendix B

Evolutionary Development Laboratory,
Department of Biological Sciences,
National University of Singapore,
14 Science Drive 4, Blk S2 Level 1,
Singapore 117543

email: ianchan@nus.edu.sg

22 Apr 2021

Dear Prof. Barrett and Prof. Rowe,

Through the electronic submission system, I have submitted our revision to the Research Article now renamed “Predation favours *Bicyclus anynana* butterflies with fewer forewing eyespots” by Ian Z.W. Chan, Zhe Ching Ngan, Lin Naing, Yueying Lee, V Gowri and Antónia Monteiro (MS Reference Number: RSPB-2020-2840.R1). We are grateful for this opportunity to revise our ms, and would like to thank you and the reviewers for the constructive comments provided in your email of 21 Mar 2021.

In this document, we provide our responses to the issues raised. Text in black are Editor’s or Referee’s comments and text in blue are our responses. Unless otherwise stated, all line numbers refer to the clean revised ms (i.e. where all tracked changes have been accepted).

Briefly, our main responses and changes are:

- 1) We have toned down our statements throughout the ms to be more circumspect in our conclusions, including limiting our claim regarding the importance of predation in the title to only our study species, as suggested by Referees 3 and 4;
- 2) We have reworked the Discussion to cover various issues in greater detail, including the possibility that our results can be explained by variation in either total eyespot number or eyespot distribution, as suggested by Referee 4; and
- 3) We made various parts of the ms more concise to meet the length requirements.

Thank you again for this opportunity to submit our revised ms. We hope that you find our responses satisfactory and that the ms is now acceptable for publication in your journal *Proceedings B*.

Yours sincerely,
Dr. Ian Chan
Research Fellow

Response to Editor

E-1) Three referees have read this version of your ms. One is happy with it, but the other two have raised some concerns that I believe are valid.

Thank you for facilitating this review. In this subsection, we briefly describe our responses to the main issues raised by each reviewer, which we address in full detail under the individual comments below.

E-2) The main concern in my opinion is that your results are over interpreted. There are a number of concerns. For example, even in the title it is stated that "predation explains" across the group, when you do not have the data to support this. In short you have taken a large step toward rejecting the hypothesis that degree of overlap is correlated to spot differences between wings (though see below), but that does not by default lend support to the hypothesis that predation does account for it.

We now limit our claim about the importance of predation in the title to our study species *Bicyclus anynana* and have toned down similar statements throughout the ms, for example in the Results and Discussion. This is detailed in our responses to Referee #3.

E-3) The support for predation is limited to your experiments with one species.

We now restrict our conclusions regarding predation to only our study species and state in the Discussion that future work is required to extend these conclusions to more nymphalid species, as described in our response to Referee #3.

E-4) Likewise, the referee points out that the experiments do not directly provide a selective explanation for the evolution of an uneven number of eyespots in the ventral side of forewing vs. hindwing.

After careful consideration, we agree with Referee #4 that our experimental design cannot disentangle whether our results were caused by the Spotty form having more eyespots or a more even distribution of eyespots. However, we respectfully disagree with a separate assertion by the reviewer, i.e. that Spotty forewings were not more attacked in our study, and present our reasons in our responses to Referee #4 below. Consistent with these two viewpoints, we have now reworked our Discussion partially according to the suggestions of the reviewer.

E-5) Similarly. the use of photos to assess overlap has some weaknesses.

We have added more discussion regarding the weaknesses of the wing overlap analysis, as described in our responses to Referees #3 and #4.

E-6) All of these suggest that you should be much more critical of your own results, and much more circumspect in your conclusions (including in the title). In my view this approach will strengthen, not weaken the impact of your work. Referee 4 has several additional suggestions that should improve the clarity and impact of the paper.

In line with these suggestions by Referees 3 and 4, we have now reworked much of our ms, from the title to the Discussion, captions and Supplementary Material.

Response to Referee #1

1-1) Congratulations for your interesting paper.

Thank you very much. We are very encouraged by your interest in our study.

Response to Referee #3

3-1) The author presents a very well written manuscript on the asymmetric distribution of eyespots in nymphalid butterfly wings. The work is embedded in relevant literature and the manuscript is nicely structured and well-organized. Their results of a comparative study and two experiments provide support for the hypothesis that selective predation plays a role in the evolution and maintenance of eyespot asymmetry (between fore- and hindwings). I have read the current version of the manuscript in detail and gone over the authors responses to reviewers comments. I believe that the authors have satisfactorily addressed all comments. I agree that the results of this study are interesting and important, and they deserve to be published.

Thank you for these encouraging comments.

3-2) Having said this, I think statements such as 'predation explains' in the title, and 'predation is the primary driver behind the asymmetric eyespot distribution observed in *B. anynana* and most other nymphalid species' in the Discussion are a not justified by the data and should be tuned-down. The work indeed reveals that predation may play a role (and this is a valuable contribution in its own right), but more data is needed (e.g. confirmation from other species and other predators, etc) to validate that predation is in fact the 'primary driver'.

We have now re-worded the title to limit the claim on predation therein to our study species *Bicyclus anynana*. We have also edited various parts of the text to tone down our statements and to be more careful when talking about extrapolating our results to other species.

Edited Text	Original Text
Lines 1-2 (Title): 'Predation favours Bicyclus anynana butterflies with fewer forewing eyespots'	'Predation explains asymmetric eyespot distribution across wings of nymphalid butterflies'
Lines 320-323: 'Our data indicate that predation helps to maintain lower forewing eyespot numbers in B. anynana because carrying more forewing eyespots draws attacks towards the forewing which is more important for flight.'	'Our data indicate that eyespot location is important and suggest that predation is the primary driver behind the asymmetric eyespot distribution observed in B. anynana and most nymphalid species.'
Lines 325-332: 'Overall, our data show that both wing overlap and predation pressure contribute to limiting the number of ventral forewing eyespots in butterflies. However, wing overlap is a limiting factor in only 12.8% of the 275 species analysed. [...] This does not immediately indicate that predation is primarily responsible for fewer forewing eyespots, but our data suggest that it might be an important factor.'	Overall, our data suggest that both wing overlap and predation pressure contribute to limiting the number of ventral forewing eyespots in butterflies. However, wing overlap is a limiting factor in only 12.8% of the 275 species analysed, indicating that predation pressure is more important in a wider range of species.

Line 400-401: ‘To extrapolate our conclusions to the Nymphalidae as a whole, similar work on other species is necessary.’	‘The extrapolation of our conclusions to the Nymphalidae as a whole would be more strongly supported if future studies produce similar results in other species.’
---	---

3-3) As a more general comment, I believe that the comparative study as conducted here, using few images from field guides, excluding intraspecific variation and phylogenetic effects, is far from ideal. Nymphalidae demonstrate a plethora of resting behaviours, and even within a single Mycalesina species there is ample variation in the relative exposure of the forewing (and even the display of dorsal vs ventral). *B. anynana*, for example, only 'folds away' their forewing after resting for a while (and this state will not often be presented in field guides because it is less informative from a taxonomical perspective). Nevertheless, I realize these data are very hard to come by, and I am confident that the observed pattern (i.e. wing overlap is a not a limiting factor) will hold.

We agree with the reviewer on the limitations of this part of our study (e.g. that there can be variation within a species and even within individuals over time). However, because of the paucity of this type of data, as the reviewer mentions, our efforts to mitigate the effects of this variation were focused on applying the following guidelines in our analysis:

- 1) First, we surveyed as many images as possible per species (ranging from one to ten, but typically two to four) and chose the most common degree of overlap observed;
- 2) Second, in cases where there was a tie, we chose the more conservative observation (i.e. where more sectors are covered).

We acknowledge that it is likely that the data will change as more images become available in future. However, because our estimate here of the number of forewing sectors exposed at rest in these 431 species is deliberately conservative, we concur with the reviewer in being confident that our conclusion is sound.

These points above are already described in our M&Ms (lines 97-101), and we have now added some extra text to the Discussion.

Additional Text
Lines 327-330: ‘Due to the paucity of this type of data, it is likely that we did not capture all the resting positions displayed by these species in nature. However, because our analysis was deliberately conservative, we believe that, even as more images become available, our conclusions should remain sound.’

3-4) For the arena experiment, giant Asian mantis nymphs *Hierodula* sp. were trapped IN a park along Upper Serangoon Road, Singapore. THEY were reared to adulthood [in the laboratory] AND only female mantids were used.

This has been edited as suggested.

Edited Text	Original Text
----------------------

Lines 120-122: ‘For the arena experiment, giant Asian mantis nymphs *Hierodula* sp. were trapped in a park along Upper Serangoon Road, Singapore. They were reared to adulthood in the laboratory and only female mantids were used.’

‘For the arena experiment, giant Asian mantis nymphs *Hierodula* sp. were trapped from a park along Upper Serangoon Road, Singapore and reared to adulthood when only female mantids were used.’

Response to Referee #4

4-1) This manuscript aims at exploring the evolutionary forces involved in the uneven number of eyespots in the forewing vs. hindwing observed in many Nymphalid butterflies. First, the degree of wing overlap on the ventral forewing pattern of 275 species is explored and the authors conclude that the reduced number of eyespots on the ventral side of the forewing probably not rely on the lack of positive selection due to wing overlap preventing selection by visual predators.

The manuscript then also described two interesting experiments with mantid predators, comparing the wing areas attacked, as well as the fitness (survival and fecundity) of *B. anynana* butterflies differing in the number of forewing eyespots. Butterflies with increased number of eyespots (spotty mutants) suffers from increased attack rate and reduced fecundity, but the experiments exploring the mantid behaviour reveals that the increase number of eyespots in the forewing does not result in increased number of attacks in the forewing but rather in an increased attack number on both wings. These experiments thus do not directly provide a selective explanation for the evolution of uneven number of eyespots in the ventral side of forewing vs. hindwing. The main caveats being that the experiments rely on the comparison of two colour-patterns differing not only in the FW/HW distribution of eyespots but also in their total number of eyespots.

Nevertheless, the reported experiments bring new insights on the predation induced by eyespots located on the ventral side of butterflies, I would thus recommend bringing clarifications on the hypotheses tested and general conclusions brought by the experiments.

Thank you for these insightful comments. We address the various issues mentioned above as they arise in the specific comments below.

Specific comments:

4-2) Title and throughout the text: the term ‘asymmetry’ to refer to difference between the forewing and the hindwing is misleading. Asymmetry would rather be appropriate to compare right and left wings. I would recommend using more precise term such as ‘uneven eyespots distribution between fore and hindwings’.

We have edited the title and other portions of the text where ‘asymmetry’ is used, as suggested.

Edited Text	Original Text
Lines 1-2 Title: ‘Predation favours Bicyclus anynana butterflies with fewer forewing eyespots’	‘Predation explains asymmetric eyespot distribution across wings of nymphalid butterflies’
Lines 17-18: ‘...but the reasons for this uneven distribution remain unclear.’	‘...but the reasons for this asymmetry remain unclear.’
Lines 21-22: ‘A second explanation is that having fewer forewing eyespots confers a selective advantage against predators.’	‘A second explanation is that the asymmetry confers a selective advantage against predators.’

Lines 29-31: ‘These results suggest that forewing eyespot number in B. anynana is limited by predation pressure: carrying more forewing eyespots is detrimental because forewings are more important for flight.’	‘These results suggest that the asymmetric eyespot distribution pattern observed across nymphalids is maintained primarily by predation pressure: a more even distribution is detrimental, possibly because flight performance is more severely hindered when forewings as opposed to hindwings are damaged.’
Line 57: ‘There are two potential explanations for this pattern...’	‘There are two potential explanations for this asymmetry.’
Line 66: ‘Second, the lower forewing eyespot number ...’	‘Second, the asymmetric distribution...’
Lines 71-72: ‘To investigate whether the pattern is maintained by predation pressure, ...’	‘To investigate whether the asymmetry is maintained by predation pressure, ...’
Lines 243-245: ‘Taken together, these three observations suggest that the lower eyespot number on forewings versus hindwings ...’	‘Taken together, these three observations suggest that the asymmetric eyespot distribution between forewings and hindwings...’
Lines 317-318: ‘Here, we find that this pattern on the ventral wing surfaces...’	‘Here, we find that this asymmetric eyespot distribution on the ventral wing surfaces...’
Lines 320-323: ‘Our data indicate that predation helps to maintain lower forewing eyespot numbers in B. anynana because carrying more forewing eyespots draws attacks towards the forewing which is more important for flight.’	‘Our data indicate that eyespot location is important and suggest that predation is the primary driver behind the asymmetric eyespot distribution observed...’
Lines 384-385: ‘...and more experiments are necessary to evaluate how variation in eyespot distribution may affect sexual signalling in nymphalids’	‘...and more experiments are necessary to evaluate how disturbances to sexual signalling, via the removal of forewing eyespots, may contribute to the eyespot distribution asymmetry observed across nymphalids.’

Abstract

4-3) Line 17: ‘Eyespots are anti-predator defence in many butterflies’: this general statement is misleading, although the eyespots have been shown to deflect attacks in some cases, the effect of eyespots on predator behaviour is not straightforward and remains to be investigated, and this study contributes to this understanding, I would avoid such over-general and vague sentence and rather start with the evolutionary question investigated in this study.

We have now removed this line.

4-4) Line 17-18: It is also misleading to report the result of a previous study in the abstract.

Because our study builds directly on the results of this previous one (Tokita et al. 2013), we feel that it is important to mention in the Abstract. However, we have now edited the text so that it appears as dissimilar to reported results as possible. Together with its position at the first line of the Abstract, this should make it clear that the text refers to a previous study.

Edited Text	Original Text
Lines 17-18: ‘There are fewer eyespots on the forewings versus hindwings of nymphalids but the reasons for this uneven distribution remain unclear.’	‘A comparative study of nymphalids found markedly fewer eyespots on forewings than hindwings but the reasons for this asymmetry remain unclear.’

References

Tokita CK, Oliver JC, Monteiro A. 2013. A survey of eyespot sexual dimorphism across nymphalid butterflies. *Int. J. Evol. Biol.* 2013, 926702. (doi: 10.1155/2013/926702)

4-5) Line 22: 275 nymphalid species ALL EXHIBITING EYESPOTS

This has now been added.

Edited Text	Original Text
Lines 22-23: ‘We analysed wing overlap at rest in 275 nymphalid species which exhibit eyespots...’	‘We analysed wing overlap at rest in 275 nymphalid species...’

4-6) Line 23: ‘does not limit’ is unclear. Did you mean that covered areas do display eyespots in many species? Here, and throughout the manuscript, it is a bit unclear why and how would the coverage of the forewing by the hindwing would prevent the evolution of eyespots ? Is it because, those partially covered eyespot are less likely to be seen by predators, and therefore less likely to be under positive selection ? This needs to be more explicit in the abstract and in the introduction.

The reviewer’s final interpretation is correct: we mean to say that eyespots on covered wing sectors would not be visible (or at best be only partially visible at times) to predators and are hence less likely to be under positive selection. We have now added text to the Abstract and Introduction to make this clearer.

Additional Text	
Lines 20-21 (Abstract): ‘...(covered eyespots are not continuously-visible and are less likely to be under positive selection)...’	
Edited Text	Original Text
Lines 22-25 (Abstract): ‘We analysed wing overlap at rest in 275 nymphalid species which exhibit eyespots and found that many have exposed forewing sectors that do not carry eyespots: i.e. wing overlap does not	‘We analysed wing overlap at rest in 275 nymphalid species and found that exposed wing area does not limit forewing ventral eyespot number in most species’

constrain the forewing from having more eyespots than the hindwing.’	
Lines 59-62 (Introduction): ‘Any eyespot on covered wing sectors would not be continuously visible to predators [14] and this wing overlap may therefore limit the number of forewing ventral eyespots that are maintained through positive selection (e.g. with a function in attack deflection).’	‘Any eyespot located on the covered parts would not be able to function as a continuously-visible signal [14] and this wing overlap may therefore limit the number of useful forewing ventral eyespots.’

4-7) Line 24: The rationale behind the two experiments should be provided here and in the introduction as well, one experiment allows to compare the fitness associated with spotty vs. wild types phenotype, while the other experiment focuses on predator behaviour and the targeted regions of wings.

We have now edited the text to state the rationale behind these two experiments more explicitly in the Abstract, Introduction and at the beginning of the two relevant subsections in the M&Ms (as suggested in comments 4-17 and 4-27).

Edited Text	Original Text
Lines 25-27 (Abstract): ‘We performed two predation experiments with mantids to compare the relative fitness of and attack damage patterns on two forms of Bicyclus anynana butterflies...’	‘We also performed two predation experiments where mantids preyed upon two forms of Bicyclus anynana butterflies...’
Lines 86-90 (Introduction): ‘We tested this by exposing Spotty and Wt forms of B. anynana to mantid predators in two experiments: (i) a microcosm experiment to quantify the effects of mantid predation on the fitness of the two forms in terms of survival, fecundity and wing damage; and (ii) an arena-based predation experiment to observe which parts of the wings are targeted in mantid attacks.’	‘We tested this by exposing Spotty and Wt forms of B. anynana to mantid predators in two different experiments: (i) a microcosm experiment to quantify the effects of mantid predation on the two forms; and (ii) an arena-based predation experiment to observe mantid attack behaviour.’
Additional Text	
Lines 129-130 (M&Ms): ‘We performed a microcosm experiment to investigate differences in survival, fecundity and wing damage in Wt and Spotty butterflies.’	
Lines 184-185 (M&Ms): ‘We performed an arena experiment to investigate differences in the wing areas targeted by mantids in Wt and Spotty butterflies.’	

4-8a) Line 28-30: This interpretation of the results is not entirely correct: butterflies with increased number of eyespots (spotty mutants) suffers from increased attack rate and reduced fecundity, but the experiments exploring the mantid behaviour reveals that the increase number of eyespots in the forewing does not result in increased number of attacks in the forewing but rather in an increased attack number on both wings.

We respectfully disagree that there was no increase in attacks on the forewings in Spotty. The arena and microcosm experiments respectively show that Spotty forewings were more attacked and more damaged than Wt forewings:

- 1) In the arena experiment, because forewings are also targeted in attacks on both wings, attacks on both wings should also be included when tallying up the total number of attacks on the forewing. This would mean that Spotty forewings were attacked more frequently—in 67.6% of Spotty trials (38.2% on both wings + 29.4% on the forewing alone) versus 57.1% of Wt trials (17.1% + 40.0%) (see Figure 1 at the bottom of this response document). We acknowledge that this was not clear in our ms and have now added text to make it clearer.
- 2) In the microcosm experiment, Spotty suffered more damage than Wt on their forewings: 0.22 damage marks per butterfly per day in Spotty versus 0.14 in Wt (the model with butterfly form explaining the data significantly better, $P = 0.04$). Although Spotty was also more damaged on the hindwings (0.17 versus 0.10 marks), this difference was not significant ($P = 0.19$). We have now edited the text to make this clearer.

Additional Text	
Lines 295-298: ‘Focusing on the forewings: because the forewing is also targeted in attacks on both wings, the forewing was attacked more frequently in Spotty (38.2% on both wings + 29.4% on the forewing alone = 67.6% of trials) than in Wt (17.1% + 40.0% = 57.1% of trials) (figure 3c).’	
Lines 335-338: ‘Spotty forewings both suffered more damage in the microcosm experiments (0.22 marks per butterfly per day versus 0.14 in Wt) and were attacked more frequently in the arena experiments (67.6% of Spotty trials versus 57.1% of Wt trials).’	
Edited Text	Original Text
Lines 285-286: ‘Together, these results indicate that Spotty experienced more intense predation and were attacked more on their forewings compared to Wt.’	‘Together, these results indicate that Spotty butterflies experienced more intense predation and were attacked at different locations on their wings compared to Wt individuals.’

In addition, the only significant difference between Spotty and Wt in the arena experiment involved the predators switching away from attacking the hindwing alone to attacking both wings simultaneously in Spotty. This result suggests that the two extra forewing eyespots in Spotty drew attacks from the hindwings forward to target both wings instead, and that the forewing became more important as a target. We have also added text to make this clearer.

Edited Text	Original Text
Lines 302-306: ‘The pairwise comparisons showed (figure 3c): (i) a significant difference in the relative proportion between “hindwing eyespots” (reducing steeply from 17.1% in Wt to 2.9% in Spotty) and “eyesspots on both wings” (more than doubling from 17.1% in Wt to 38.2% in Spotty) ($P = 0.031$),	‘The pairwise comparisons showed (figure 3c): (i) a significant difference in the relative proportion between “hindwing eyespots” (reducing steeply from 17.1% in Wt to 2.9% in Spotty) and “eyesspots on both wings” (more than doubling from 17.1% in Wt to 38.2% in Spotty) ($P = 0.031$); ...’

suggesting that the mantids shifted from attacking the hindwings alone to attacking both wings simultaneously; ...’	
Lines 310-311: ‘These data suggest that the predominant change in mantid attacks was a switch from targeting the hindwing in Wt to attacking both wings in Spotty.’	‘These data suggest that the mantids switched from striking the hindwing eyespots alone and (to a lesser extent) the forewing eyespots alone in Wt to attacking eyespots on both wings simultaneously in Spotty butterflies.’
Lines 599-602 (Figure 3 caption): ‘Comparing Spotty to wildtype, the results suggest that the mantids predominantly switched away from striking the hindwing eyespots alone in wildtype (this category is much reduced in Spotty) to attacking the eyespots on both wings in Spotty (this category more than doubled to almost 40% in Spotty).’	‘Comparing Spotty to wildtype, the results suggest that the mantids switched away from striking the forewing eyespots alone and hindwing eyespots alone in wildtype (these two categories, especially the hindwing, are much reduced in Spotty) to attacking the eyespots on both wings in Spotty (this category more than doubled to almost 40% in Spotty).’
Additional Text	
Lines 354-359: ‘In the arena experiment, the additional forewing eyespots in Spotty caused a significant reduction of attacks towards hindwings and an increase towards both wings, suggesting that the forewing became a relatively more important target. We believe our experiments shed light on the detrimental effect of drawing too much attention to forewings by placing too many eyespots on these wings, relative to placing most eyespots on the hindwings.’	

4-8b) These experiments thus do not directly provide a selective explanation for the evolution of uneven number of eyespots in the ventral side of forewing vs. hindwing. The main caveats being that the experiments rely on the comparison of two colour-patterns differing not only in the FW/HW distribution of eyespots but also in their total number of eyespots. The experiments clearly show that individuals with more eyespots suffers from increase attacks, but does not allow to conclude about the selection of uneven number in FW/HW.

There are two reasons for our current experimental design. In order to maintain a constant number of total eyespots between our two forms, as suggested, we would have to remove two hindwing eyespots for the two forewing eyespots that were added with the Spotty variant, e.g. the control (Wt) form would have 2:7 forewing:hindwing eyespots and the experimental form would need to have 4:5. This would introduce an additional difference between the forms: the form with the more uneven forewing-hindwing eyespot distribution (i.e. Wt) would also have more hindwing eyespots (i.e. 7 in Wt compared to 5 in Spotty). It is hard to say whether this is an improvement over our current design, but it is something that we agree should be attempted in future, and we have now added text to the Discussion to point this out (lines 359-363). However, given the main objective of these predation experiments (i.e. to demonstrate the fitness benefits of having fewer forewing eyespots in *B. anynana*) and the limitations of the currently available mutant lines, we decided to keep the number of hindwing eyespots constant and varied only the number of forewing eyespots.

After careful consideration, we do agree with the reviewer that, from *our data alone*, we cannot distinguish whether the higher attack rates observed in Spotty were a consequence of having more eyespots overall or a more even distribution of eyespots. We can only say that the increase in forewing eyespot number in Spotty caused it to suffer more damage (overall, but especially on the forewing, as we describe in our previous response, 4-8a) and to be less fit than Wt. We have therefore now revised or removed text throughout the ms which explicitly claimed that our results explain the uneven eyespot distribution.

Edited Text	Original Text
[Removed from Discussion]	‘These results indicate that the mantids considered the distribution of eyespots across both wings in their attack decision as opposed to eyespot number on the forewing alone.’
Lines 371-373: ‘A greater number of forewing eyespots, as seen in Spotty, may therefore be selectively disadvantageous because this encourages predators to grasp both wings simultaneously.’	‘A more even distribution of eyespots across fore and hindwings, as seen in Spotty, may therefore be selectively disadvantageous because this encourages predators to grasp both wings simultaneously.’

We believe, however, that when *informed by the results of Prudic et al. (2015)*, our data suggest that eyespot location is important. Previous work by Prudic et al. (2015) showed that the more conspicuous wet season (WS) form of *B. anynana*, with large and clear eyespots, was primarily targeted on their hindwings (by a different species of mantis) whereas the less conspicuous dry season (DS) form, with small and faded eyespots, was attacked on the head and thorax. The WS form had a much higher level of wing damage overall, but also had higher fitness. Hence, if Spotty were more attacked simply because they were more conspicuous (having more eyespots overall), we would expect more wing damage but not necessarily lower fitness. However, we observed both in the microcosm experiment. This suggests to us that the lower fitness of Spotty observed in our experiments is due to the higher level of forewing damage relative to hindwing damage. In our arena experiment we clearly saw how the presence of additional eyespots on the forewing, which made this wing more conspicuous, led to a change in the location of first mantid attacks: there was a significant reduction of attacks towards hindwings and an increase in attacks towards both wings (in Spotty). This means that the forewing became a relatively more important target for the attacks. Nymphalids display large variation in total number of eyespots but they consistently have fewer eyespots on their forewings (Tokita et al. 2013). We believe our experiments shed light on the detrimental effect of drawing too much attention to forewings by placing too many eyespots on these wings, relative to placing most eyespots on the hindwings. Nevertheless, we agree with the reviewer that a future experiment is needed to compare forms of the same butterfly genotype with different eyespot distributions whilst keeping total eyespot number across both wings constant. We have now added text to discuss these ideas.

Additional Text
Lines 344-363: ‘Although we show that having more forewing eyespots is detrimental, our results alone cannot disentangle whether this a consequence of Spotty having a more even

distribution of eyespots (across forewings and hindwings) or more eyespots overall. However, when informed by existing studies, our data suggest that eyespot location may be important. Prudic et al. [7] showed that, although the conspicuous wet season form (WS) of *B. anynana* was more attacked than the less conspicuous dry season form (DS), the WS still had higher fitness (because most damage was on the hindwings compared to the head and thorax in DS). Hence, if Spotty were more attacked simply because they were more conspicuous (having more eyespots overall), we would expect more wing damage but not necessarily lower fitness. However, we observed both in the microcosm experiment. We therefore believe that the lower fitness in Spotty is due to greater forewing relative to hindwing damage. In the arena experiment, the additional forewing eyespots in Spotty caused a significant reduction of attacks towards hindwings and an increase towards both wings, suggesting that the forewing became a relatively more important target. Nymphalids display large variation in total eyespot number but consistently have fewer forewing eyespots [13]. We believe our experiments shed light on the detrimental effect of drawing too much attention to forewings by placing too many eyespots on these wings, relative to placing most eyespots on hindwings. Nevertheless, future studies are needed to differentiate between the effects of total eyespot number and eyespot distribution, e.g. by painting eyespots onto the same butterfly genotype to produce forms with different eyespot distributions but the same total eyespot number.’

References

Prudic KL, Stoehr AM, Wasik BW, Monteiro A. 2015. Eyespots deflect predator attack increasing fitness and promoting the evolution of phenotypic plasticity. Proc. R. Soc. B 282(1798), 20141531. (doi: 10.1098/rspb.2014.1531)

Tokita CK, Oliver JC, Monteiro A. 2013. A survey of eyespot sexual dimorphism across nymphalid butterflies. Int. J. Evol. Biol. 2013, 926702. (doi: 10.1155/2013/926702)

Introduction:

4-9) Line 40: the term ‘function’ sounds finalistic and is often misleading, because most trait evolve under multiple selective pressures. The evolution of eyespots can result from trade-offs between conspicuousness and deflection for instance, so that this sentence should be entirely rephrased to explain the selective pressures at play and highlight that their relative contribution remains to be studied.

We have rephrased the sentence to focus on eyespot effectiveness in interactions with predators.

Edited Text	Original Text
Lines 40-43: ‘Eyespot effectiveness is dependent on multiple factors, e.g. eyespot size and number [6,7], behaviour such as rapid wing displays or stridulation [8], and predator type, as the same eyespot may appear intimidating to some predators but not to others [5].’	‘Which of these functions plays the dominant role, however, is dependent on a variety of factors, e.g. the size and number of eyespots on the wing [6,7], associated behaviour such as rapid wing displays or stridulation [8], and the type of predator, as the same eyespot may appear large and intimidating to some predators but small and non-intimidating to others [5].’

4-10) Line 47: replace ‘the eyespots’ attack deflection function’ by ‘deflection of the attacks on the eyespots, located in less-vital parts of the body’.

This has been edited with minor differences to the suggested phrasing.

Edited Text	Original Text
Lines 45-47: ‘Clusters of small eyespots on butterfly wing margins appear to make the area more conspicuous to predators and thereby deflect attacks onto themselves, away from vital body parts.’	‘Clusters of multiple small eyespots on butterfly wing margins appear to make the area more conspicuous to predators and thereby enhance the eyespots’ attack deflection function.’

4-11) Line 58: removed ‘it is interesting that’.

This has now been rephrased.

Edited Text	Original Text
Line 55: ‘Surveying museum specimens of 451 nymphalid species...’	‘It is interesting that, in a survey of 451 nymphalid species...’

4-12) You may also detail why you focus on the ventral side of the wings and do not investigate the effect coverage of wing on the evolution of eyespots on the dorsal side as well.

Because most nymphalids rest with their wings folded, dorsal eyespots are generally less involved in anti-predator defence and play a greater role in sexual signalling.

Additional Text
Lines 57-58: ‘...especially with respect to ventral eyespots which are important in anti-predator defence.’

4-13) Line 63: replace ‘be able to function as a continuously-visible signal ’ by ‘be continuously-visible’

This has now been edited.

Edited Text	Original Text
Lines 59-60: ‘Any eyespot on covered wing sectors would not be continuously visible to predators...’	‘Any eyespot located on the covered parts would not be able to function as a continuously-visible signal...’

4-14) Line 64: the term ‘useful’ is definitely finalistic, and this misleading sentence prevent a proper description of the selective force involved: did you mean prevent positive selection due to attack deflection?

The reviewer’s interpretation is correct. We have now edited the text to make our meaning clearer.

Edited Text	Original Text
Lines 60-62: ‘...and this wing overlap may therefore limit the number of forewing ventral eyespots that are maintained through positive selection (e.g. with a function in attack deflection).’	‘...and this wing overlap may therefore limit the number of useful forewing ventral eyespots.’

4-15) The deimatic effect, although poorly studied should probably not be neglected, and is likely to depend on the relative size and shape of the different eyespots, that has not been investigated here. It is particularly surprising to discard this hypothesis so quickly, because as shown on fig.1A, the biggest and most conspicuous eyespot observed on the ventral side of the wings of *B. anynana* is located on the forewing, and in a partially covered area.

We agree that forewing eyespots covered by the hindwing could indeed be used for deimatic behaviour. This topic is under-studied, and it would be important to examine questions such as the shape and size of eyespots which are covered versus exposed, and the proportion of species with covered eyespots. This is an interesting potential application of our dataset and we have now added text to the Discussion calling for future research.

Additional Text
Lines 398-400: ‘Deimatic displays, which have received limited study, could also potentially influence eyespot shape, size and distribution, and are an interesting avenue for further research.’

However, we have not observed deimatic behaviour in *B. anynana* (and now mention this in the M&Ms). The specific pressures acting on covered eyespots are likely to differ from those acting on exposed ones and we do not examine deimatic behaviour in this study. We instead restrict ourselves here to the specific question of whether the number of exposed forewing sectors limits nymphalids in general to having fewer exposed forewing than hindwing eyespots, i.e. if a butterfly has fewer exposed forewing sectors than hindwing sectors carrying eyespots, it cannot have more exposed forewing eyespots than exposed hindwing eyespots. We have now added or edited text in the Introduction and M&Ms to make our hypotheses clearer (as suggested in comment 4-32).

Additional Text	
Lines 62-64 (Introduction): ‘In other words, if a butterfly has fewer exposed forewing sectors than hindwing sectors carrying eyespots, it cannot have more exposed forewing eyespots than hindwing eyespots.’	
Edited Text	Original Text
Lines 65-66: ‘... has not been documented in our study species B. anynana .’	[Deimatic behaviour] ‘...has not been widely documented for ventral forewing eyespots.’
Lines 103-115 (M&Ms): ‘We then investigated whether exposed wing area limits forewing eyespot number to being	‘We then investigated whether exposed wing area limits the number of forewing eyespots through three analyses.’

less than hindwing eyespot number through three analyses.’

‘First, in all species with eyespots, we divided the number of eyespots by the number of exposed sectors on each wing to investigate whether there are still fewer forewing versus hindwing eyespots after controlling for the number of exposed wing sectors.’

‘Second, we compared the number of exposed forewing versus hindwing eyespots in only those species which are not constrained to having fewer exposed forewing eyespots by wing overlap, i.e. these species have equal or more exposed forewing sectors (either with or without eyespots) than hindwing eyespots.’

‘Third, in all species with eyespots, we tested whether the absence of the M2 and M3 eyespots on the forewing (which differentiate the two *B. anynana* forms in our study), when homologs are present on the hindwing, can be accounted for by the frequency that these forewing sectors are covered. Assuming similar developmental constraints between the forewing and hindwing, any difference between the eyespot distributions would be a result of selection.’

‘First, we divided the number of eyespots by the number of exposed wing sectors on each wing to control for the number of exposed sectors directly.’

‘Second, in species which have more exposed forewing sectors than hindwing eyespots (and which are therefore not limited to having fewer exposed forewing than hindwing eyespots), we compared the number of exposed forewing versus hindwing eyespots.’

‘Third, focusing on the forewing M2 and M3 sectors—the specific sectors on which the two forms of *B. anynana* in our study differ—we tested whether the absence of these eyespots on the forewing, when homologs are present on the hindwing, can be accounted for by the frequency that these forewing sectors are covered by the hindwing.’

4-16) Line 67 and 73: As mentioned above, the term ‘asymmetry’ is not entirely appropriate here.

We have replaced ‘asymmetry’ here and throughout the ms, as described in response 4-2 above.

4-17) Line 92-95: The questions investigated in the two experiments should be more explicit: the microcosm experiments allow to estimate the effect of the two phenotypes on the fitness of the individuals by comparing survival and reproductive success with and without predators, while the area experiments precisely investigate the region of the wings targeted by predator’s attacks, and therefore the selection acting on the distribution of eyespots.

We now describe the questions more explicitly. Please refer to response 4-7 above.

4-18) I would also advise to present first the behavioural experiment and then the impact on fitness.

Thank you for this suggestion. We tried both alternatives while writing up the ms but eventually decided that we prefer the current order because the microcosm experiment gives data on both

fitness and wing damage, which we can then support with the observed attack behaviour of the predators.

Materials and Methods

4-19) Line 104-105: Is there a reason for ignoring behaviour variation within species? The mean position covered could be used? Would it change the results or is this variation observed for a large or a small proportion of the species ? These data could be of interests for a large number of entomologists, the authors indeed provide a Dryad number, but I was not able to access it and therefore could not check them (maybe because the data will be online only after the publication ?).

Variation was observed in approximately 40% of species. We handled the variation by: (i) sampling as many images as possible for each species and using the most frequent observation (i.e. the mode); and (ii) when where there was a tie, using the most conservative observation, i.e. with the most forewing sectors covered (lines 97-101 in the M&Ms). Usage of the mean instead would not change the overall conclusion but might lend stronger support to our results because our current estimate in most cases is based on the most conservative observation. However, we recognise that this type of data is very scarce, and it is almost certain that we did not observe all the resting positions displayed by all the species in nature. We acknowledge the importance of this point and have now added text to the ms to discuss it.

Additional Text

Lines 327-330: 'Due to the paucity of this type of data, it is likely that we did not capture all the resting positions displayed by these species in nature. However, because our analysis was deliberately conservative, we believe that, even as more images become available, our conclusions should remain sound.'

The data was uploaded to Dryad as per the requirements of the journal. I recall that the link is made accessible to the public only after publication, but I am uncertain about whether or how the journal makes the link accessible to reviewers. In the interim, we have now also submitted the dataset through the online system as part of the Supplementary Material for the reviewers to access.

4-20) Line 123-125: Is there any reason to use different predators in the two experiments ? And to use female only in the arena experiments while using both sex in the microcosm. As mentioned in the introduction, both are probably relevant predators, but it makes the two experiments less related to each other.

This was due to logistical constraints: we were not able to procure the same species for both experiments and, for the arena experiments, we could not trap sufficient males to maintain a balanced design.

4-21) Line 126-127: Did the wild-type and mutant shared the same genetic background ? Are spotty butterflies related to each other ? An increase relatedness among individuals could affect their fertility within cage.

The Spotty mutation emerged from within the wildtype line in the lab in the 1990s and has been bred as a separate large colony ever since. We have detected no symptoms of inbreeding depression in our Spotty stock and our data show no differences in fertility between Wt and Spotty in our microcosm experiments in the cages without mantids.

4-22) Line 143-144: Was wing damage quantified in both control and experimental cages? Did the butterflies have wing damage in cage without predators (because they rub their wing against the cage walls for instance). I guess Fig. 2C reports damage in cages with predators? It would be important to disentangle the effect of the damages cause by predators from the damages due to aging and rubbing against the cages' walls.

We did measure damage on the butterflies' wings in control cages (we realise that this was not made clear and have now amended our text in the M&Ms, lines 136-137). There was no difference between the damage patterns on Wt and Spotty butterflies in control cages. Comparing control to experimental cages overall, individuals in control cages showed more damage on the leading (i.e. costal and apical) edges of their forewings but less damage everywhere else (Figure S2). We observed that this damage accumulated slowly over time, only starting to become more evident after one week. This period is much longer than the average survival of Spotty butterflies in experimental cages (3.4 days) and similar to the average survival of Wt butterflies (9.4 days).

This potentially explains why Wt butterflies have more damage than Spotty on the leading edge of their forewings: they survive longer and accumulate more damage associated with wear and tear, which Figure S2 shows to occur at precisely these areas. We now mention this in the caption of Figure S2).

Additional Text

Lines 34-37 (Supplementary Material, Figure S2 caption): 'This also potentially explains why, in our experimental cages, Wt butterflies (which survived longer than Spotty and therefore would accumulate more damage associated with wear and tear) have relatively more wing damage on the leading edges of their forewings.'

We did initially attempt to correct for wear and tear but this is non-trivial as the damage does not accumulate linearly, i.e. we cannot simply subtract average damage scores (per butterfly per day) in control cages from average damage scores in the corresponding experimental cages. We note, however, that in spite of Wt butterflies surviving longer (and therefore accumulating more wear and tear than Spotty), we still observe that Spotty wings are more damaged. Therefore, correcting for wear and tear would likely result in even stronger support for our current

conclusions. We acknowledge that this is important to mention and have now added it to the Discussion.

Additional Text

Lines 338-342: ‘Whilst we did not explicitly control for wing damage due to wear and tear in our microcosm experimental cages, our data show that this accumulates over time primarily on the leading edge of the forewing (Figure S2). Therefore, correcting for wear and tear would result in a greater reduction of damage scores on the forewings of Wt (which survived longer than Spotty), which would provide stronger support for our conclusions.’

4-23) Line 147-148: The rationale behind the use of the 0.5 scoring is not entirely clear. Here, it is hard to assess whether a mark spreading two wing cells was due to a single attack or a succession of several attack and subsequent wing rubbing. It would probably be more conservative, to have a simpler binomial score for each wing cell describing whether it was intact or damage. For a quantitative measure, you would need to compute, for each cell, the proportion of the wing cell surface removed.

We also performed a binomial analysis on the wing damage data (‘1’ for damaged or ‘0’ for undamaged), and the results were similar. However, we feel that the system currently presented is actually more conservative than a binomial 1 or 0 score. Assuming two sectors are damaged by the same mark, the binomial scoring system would result in a score of 1 for both sectors whereas our system would result in a score of 0.5 for both sectors to account for the fact that they were both damaged in the same attack. The score in any given sector would only be greater than 1 if there were at least two clear instances of damage (i.e. marks which are clearly separate)—information that the binomial system would not be able to capture.

4-24) Line 166: Was the number of eggs laid divided by the number of butterflies alive during each day? Wouldn’t it be more relevant to control for the number of females still alive, because only females are able to lay eggs? I am not familiar with *B. anynana*, but for other butterfly species, females do not remain virgin in a cage with males, and the number of eggs will thus strongly depend on the sex-ratio, females being frequently harassed by males. It would be relevant to control for the changes in sex-ration throughout the experiments when studying the number of eggs produced.

The total number of eggs collected from the cage was divided by the total number of butterflies initially in the cage (10). The situation described by the reviewer is possible but would only be a problem if there were systematic differences in how the sexes were targeted in the Wt and Spotty cages. Unfortunately, we did not record the sex of the surviving butterflies within the cages each day. However, we did not notice any signs of sex-biased mortality in this (and previous) studies and have no reason to expect it to occur. We do acknowledge that it is a possibility and have now added it to the Discussion as a consideration for future work.

Additional Text

Lines 385-387: ‘These studies should, however, consider possible sex-biased mortality (as sex ratio within a cage affects fecundity) and could also investigate female preference in the absence of harassment from males.’

4-25) Line 167: ‘problems with error distribution’, this must be more precise.

This has now been elaborated upon.

Edited Text	Original Text
Lines 163-164: ‘Models were checked for adherence to error distribution assumptions using the DHARMA package [30].’	‘Models were checked for problems with error distribution using the DHARMA package [30].’

4-26) Line 171: the percentages reported on the heat map (fig.2C) are not straightforward: for example, the green part on the leading edge of the FW: does it mean that twice as much wt individuals were damaged at this location with respect to spotty individuals? Is it standardized by the total number of individual damages at this location? Please provide more detail on the way these percentages were computed.

Thank you for pointing this out. The percentages refer to the average damage score calculated for that sector. A sector coloured dark green (i.e. +50% in the “Wildtype damaged more” direction) means that the average damage score for that sector was more than 50% higher in Wt than in Spotty. We have now specified this in the text and the figure caption.

Edited Text	Original Text
Lines 278-279: ‘The two-way average damage score heat map (figure 2c)...’	‘The two-way damage heat map (figure 2c)...’
Lines 587-588 (Figure 3 caption): ‘A two-way heat map with each wing sector coloured according to the form which suffered higher average damage scores in that sector (green for wildtype and yellow for Spotty; only experimental trials shown).’	‘A two-way heat map with each wing sector coloured according to the form which suffered more damage (green for wildtype and yellow for Spotty; only experimental trials shown).’

4-27) Line 188 and line 131: the general purpose of each of the two experiments should be given at the beginning of the sections, for the reader to immediately know what will be tested in each experiment.

This has now been added. Please refer to response 4-7 above.

4-28) Line 216: I did not understand why you studied the first four strikes while each mantid was tested 8 times (with four spotty and four wt butterflies, as mentioned on line 208).

All trials from all mantids were used for the dataset. By the phrase ‘four first strike categories’ (in the previously reviewed ms), we refer to the four categories of first strikes: i.e. on the body, the forewing alone, the hindwing alone, and both wings simultaneously. We acknowledge that this could be made clearer and have now edited the text and moved the description of the four categories forward to the M&Ms, as suggested in comment 4-41 below.

Additional Text	
Lines 207-210: ‘First strikes observed were assigned to one of four categories (figure 3b): (i) on or near the “body” (including the proximal parts of the wing, without any eyespots); (ii) on the “forewing eyespots”; (iii) on the “hindwing eyespots”; and (iv) on the “eyesspots of both wings” simultaneously.’	
Edited Text	Original Text
Lines 214-215: ‘...multinomial regression compares the overall proportions of the four categories of first strikes observed...’	‘...multinomial regression compares the overall proportions of the four first strike categories...’

4-29) Line 217: Does the ‘type of strike’ relate to the position targeted by the mantid (as described on line 296-300) ? If so, this should be clearly stated here.

Yes, this refers to the four categories of first strikes. We have now defined the categories earlier (i.e. in the M&Ms, as described in response 4-28 above) and edited this text to make it clear that it refers to the four categories defined.

Edited Text	Original Text
Line 216: ‘The response variable was the proportion of each first strike category...’	‘The response variable was the proportion of each type of first strike...’

4-30) Line 219: Are ‘first strike categories’ similar to the ‘type of first strike’ ? It would be easier to define those types or categories and use a consistent vocabulary.

The reviewer’s understanding is correct. We have now edited the text to consistently use ‘category’ or ‘categories’ throughout the ms.

Edited Text	Original Text
Line 218: ‘...to identify which specific categories were different.’	‘...to identify which specific locations were targeted differently.’
Lines 300-301: ‘The multinomial GLMM revealed a marginally significant difference in the overall proportions of the four first strike categories...’	‘The multinomial GLMM revealed a marginally significant difference in the overall proportions of the four first strike locations...’
Line 570 (Table 1 header): ‘Total number of first strikes in each category...’	‘Total number of first strikes on each body position...’

Results

4-31) Line 235: ‘Wing overlap is a limiting factor ‘: this is an interpretation, Does it mean that in 12.8% of species, no eyespot were observed in the FW zone covered by the HW ? This should be precisely described. Similarly, on line 238, it is rather difficult to understand what is the result for the 240 remaining species ? Do they display eyespot in the FW zone covered by the HW ?

We mean to say that, in 12.8% of the species, the number of exposed forewing sectors is less than the number of hindwing eyespots so that it is not possible for the butterfly to have as many exposed forewing eyespots as hindwing eyespots. Our hypothesis is that butterflies have fewer forewing eyespots because this increases their fitness but, in these 12.8% of species, we cannot be sure whether they have fewer exposed forewing eyespots because it is beneficial or because there are not enough exposed forewing sectors to carry eyespots. We therefore remove these 12.8% of species from our comparison, and compare the number of forewing versus hindwing eyespots in the remaining 240 species (which have the potential to have more exposed forewing eyespots than hindwing eyespots because they equal or more exposed forewing sectors than hindwing sectors with eyespots).

We have amended the text to make these ideas clearer.

Edited Text	Original Text
Lines 232-239: ‘Second, wing overlap is a limiting factor for the number of forewing eyespots in only 35 species (12.8%). These species have fewer exposed forewing sectors than hindwing sectors with eyespots and therefore cannot have as many exposed forewing eyespots as hindwing eyespots. The remaining 240 species (87.2%), including Bicyclus anynana, have the potential to display the same number or more exposed forewing than hindwing eyespots and yet they have, on average, more than two times as many exposed eyespots on the hindwing (Mean \pm SE: 4.0 ± 0.14) than the forewing (1.8 ± 0.12), suggesting that this difference was not caused by wing overlap.’	‘Second, wing overlap is a limiting factor for the number of forewing eyespots in only 35 species (12.8%). These species have fewer exposed forewing sectors than hindwing eyespots. In the remaining 240 species (87.2%), the number of forewing eyespots is not limited by the number of exposed wing sectors and yet they have, on average, more than two times as many exposed eyespots on the hindwing (Mean \pm SE: 4.0 ± 0.14) than the forewing (1.8 ± 0.12).’

4-32) If I understood correctly, only a limited number of species had no eyespots in the covered part of the forewing despite displaying eyespots on the rest of the wings. The authors conclude that the reduced number of eyespots on the ventral side of the forewing probably not rely on the lack of positive selection due to wing overlap preventing selection by visual predators. The hypothesis behind this survey is nevertheless not entirely clear, since eyespot can also be involved in deimatic behaviour, promoting the evolution of eyespot in wing areas covered by the other wing. A more thorough analysis including comparisons of the eyespots shapes and size in covered vs. uncovered areas of both ventral and dorsal sides of the wings would probably bring

more solid conclusions on the effect of wing overlap on the evolution of eyespot number in Nymphalid butterflies.

We agree that deimatic behaviour is potentially an important factor and warrants future research. However, it is not the focus of this study; we examine specifically whether there are generally fewer exposed forewing sectors than hindwing eyespots in nymphalids. We have now edited text throughout the ms to make our hypotheses clearer. Please refer to response 4-15 above.

4-33) Lien 242: the homology of FW and HW cells should be explained in the introduction, and assuming these common developmental constraints, explain that the working hypothesis for the difference between HW and FW distribution of eyespots is interpreted as a result of selection. This working hypothesis is not mentioned in the introduction.

We now describe this working hypothesis in the M&Ms where this analysis is first described.

Additional Text
Lines 114-115: 'Assuming similar developmental constraints between the forewing and hindwing, any difference between the eyespot distributions would be a result of selection.'

4-34) Line 244: again replace the term 'asymmetry'

We have replaced 'asymmetry' here and throughout the ms, as described in response 4-2 above.

4-35) Line 230,247,291: The title of the result subsection must explicitly relate to the hypothesis tested or the results found rather than repeating the methods used. I would probably report the behavioural experiment looking at the target of selection (arena experiment) before the estimation of the impact of phenotype on fitness (microcosm experiment).

We have now amended the subheadings.

Edited Text	Original Text
Line 228 (subheading 3a): 'Wing overlap partially explains the uneven eyespot distribution'	'Wing overlap analysis'
Lines 247-248 (subheading 3b): 'Microcosm experiment: Spotty butterflies have lower fitness and suffer more wing damage'	'Microcosm experiment'
Line 288 (subheading 3c): 'Arena experiment: Spotty and Wt butterflies are targeted differently'	'Arena experiment'

4-36) Line 265: following up with my sex-ratio survey suggested above, did butterfly females died sooner than male ?

Unfortunately, these data were not collected. However, we did not notice any signs of sex-biased mortality in this (and previous) studies and have no reason to expect it to occur. We acknowledge that it remains a possibility and have added text discussing the issue.

Additional Text

Lines 385-387: ‘These studies should, however, consider possible sex-biased mortality (as sex ratio within a cage affects fecundity) and could also investigate female preference in the absence of harassment from males.’

4-37) Line 269: Was the number of eggs computed per butterfly ALIVE ? If so, the higher mortality on spotty should not be accounted for the reduced fertility ?

The total number of butterflies initially in the cage (i.e. 10 in all cages) was used to calculate the average, so the higher mortality has not yet been factored into the computations.

4-38) Fig2C: the high attack rate on the green part on the leading edge of the FW of the wt individuals is not mentioned, did you have an interpretation for this surprising pattern ? (see also my related comment on the material and method section).

Yes, we believe this was due to increased wear-and-tear in the longer-lived Wt. Please refer to response 4-22 above where we describe this in greater detail.

4-39) Line 288-289: The results of this experiments are thus in line with the hypothesis of increased predation associated with increased number of eyespots, but do not show an increased number of damages on the FW in the spotty individuals, in line with the arena experiment.

We agree and discuss the inability of our experimental design to distinguish between the effects of having more eyespots or a more even distribution of eyespots in response 4-8b above. However, we argue that we did observe increased damage on the forewings of Spotty in response 4-8a above.

4-40) Line 295: ‘In eight OUT of the 69 attacks’

The word ‘out’ has now been added.

4-41) Line 296-300: This should be moved to the Material and Method section.

This has now been moved to the M&Ms, as detailed in response 4-28.

4-42) Line 309: This result thus contradicts the deflection hypothesis, the increased number of eyespots in the FW of spotty does not increase the attack toward the FW, this should be more explicitly stated here.

We respectfully disagree that there was no increase in attacks on the forewing (please see response 4-8a above). However, we fully agree with the reviewer that we cannot tell whether this was due to Spotty having more eyespots or a more even eyespot distribution, and have now adjusted our Discussion accordingly (please see response 4-8b above).

Discussion:

4-43) Line 318: As explained below, this general sentence is useless and misleading and should be removed.

We have now removed this sentence.

4-44) Line 321: replace ‘asymmetry’ by a more precise term.

We have replaced ‘asymmetry’ here and throughout the ms, as described in response 4-2 above.

4-45) Line 323: here phenotype with more eyespots in the FW were compared with phenotype with less eyespot in the FW but also less eyespots in total. It is thus not possible to actually conclude about the effect of uneven distribution. Moreover, in both experiments, the increased number of eyespots in the FW of spotty does not especially increase the attack toward the FW, but it does increase the attacks on both wings (arena experiments) and lead to an increase attack rates. This is in line with the hypothesis that increase number of eyespots might increase conspicuousness. This result however contradicts the deflection hypothesis, because the spotty phenotype was not more heavily attacked on the FW. This should be properly discussed and the trade-off between conspicuousness and deflection in the evolution of eyespots would be explained and discussed.

We agree with the reviewer and discuss the inability of our experimental design to distinguish between the effects of having more eyespots or a more even distribution of eyespots in response 4-8b above. However, we argue that we did observe increased damage on the forewings of Spotty in response 4-8a above.

4-46) Lines 350-351: This interpretation is not supported since only two eyespots distribution were tested, and is not related to the general question of the evolution of eyespot number and distribution on butterfly wings, this should be removed or rephrased.

This has now been removed, as described in response 4-8b above.

4-47) Lines 352-363: The hypothesis that wing damages on the FW are probably more detrimental to flight (and therefore to survival and/or reproductive success) is supported but seems a bit off-topic since spotty butterflies were not necessarily more attacked on the FW.

The microcosm and arena experiments show that Spotty butterflies were respectively more damaged and more attacked on the forewing (please see response 4-8a above), so this discussion point would still be relevant.

4-48) Lines 365-376: As mentioned below, tracking down the sex-ratio throughout the microcosm experiment would help understanding what is driving the decrease in egg-laying in the cage with spotty butterflies. Once again, I am not familiar with *Bicyclus anynana* but in many butterfly species, females struggle to escapes male in a cage, and I am not sure they are really able to express their preferences in such an experimental set-up (no choice experiment with a constant harassment from males).

We agree with the reviewer on this point and have now added these considerations to the Discussion, as described in responses 4-24 and 4-36 above.

Figures

Figure 1. Arena experiment results (extracted from Figure 3c in the ms). The forewing is more attacked in Spotty: the forewing was attacked in 57.1% of wildtype trials (40.0% on forewing eyespots only + 17.1% on both wings) and 67.6% of Spotty trials (29.4% on forewing eyespots only + 38.2% on both wings). Attacks targeting hindwing eyespots alone decreased by nearly four-fifths from wildtype to Spotty while attacks targeting both wings more than doubled—this was the only significant difference between the two forms ($P = 0.03$)—suggesting that this is the main difference in attack behaviour caused by the extra forewing eyespots in Spotty was a switch from targeting the hindwings to both wings simultaneously.

Appendix C

Evolutionary Development Laboratory,
Department of Biological Sciences,
National University of Singapore,
14 Science Drive 4, Blk S2 Level 1,
Singapore 117543

email: ianchan@nus.edu.sg

6 May 2021

Dear Prof. Barrett and Prof. Rowe,

We are delighted to receive your acceptance of our ms—“Predation favours *Bicyclus anynana* butterflies with fewer forewing eyespots” by Ian Z.W. Chan, Zhe Ching Ngan, Lin Naing, Yueying Lee, V Gowri and Antónia Monteiro (MS Reference Number: RSPB-2020-2840.R2)—for publication as a Research Article in *Proceedings B*.

Through the electronic submission system, I have submitted the revisions requested in your email of 4 May 2021. In this document, we provide our responses to the issues raised. Text in black are the Editor’s comments and text in blue are our responses. Unless otherwise stated, all line numbers refer to the clean revised ms (i.e. where all tracked changes have been accepted).

Thank you very much for this opportunity to publish in your journal *Proceedings B*.

Yours sincerely,
Dr. Ian Chan
Research Fellow

Response to Editor

I would like to thank the authors again for making such a good effort at attending to each of the comments raised by the reviewers. I think that you have done well in the text by clarifying issues relating to the attack rates on the forewings in the arena experiment and also generally being more cautious about your interpretation of the data.

Thank you very much for facilitating this review and for the detailed suggestions below.

I have a few very minor comments to make below:

L30: Sounds almost as though one of the findings was that “carrying more forewing eyespots is detrimental because forewings are more important for flight.” This is not really something that the ms investigated. I suggest making this clearer: “These results suggest that forewing eyespot number in *B. anynana* is limited by predation pressure. This may occur if attacks on forewing eyespots have more detrimental consequences for flight than attacks on hindwing eyespots.”

We have now amended the text as suggested, with minor changes.

Edited Text	Original Text
Lines 29-31: ‘These results suggest that predation pressure limits forewing eyespot number in B. anynana . This may occur if attacks on forewing eyespots have more detrimental consequences for flight than attacks on hindwing eyespots.’	‘These results suggest that forewing eyespot number in B. anynana is limited by predation pressure: carrying more forewing eyespots is detrimental because forewings are more important for flight.’

L65: Deimatic is a great word but I am not sure that most people would be familiar with its meaning. I would consider this jargon.

We have now replaced the word “deimatic” with “startle” throughout the ms.

Edited Text	Original Text
Line 64-65: ‘Although covered eyespots may function in startle displays (e.g. [15,16])...’	‘Although covered eyespots may function in startle displays (e.g. [15,16])...’
Line 399-400: ‘Startle displays, which have received limited study...’	‘Deimatic displays, which have received limited study...’

L66: Not 100% clear why: Perhaps try “Second, higher numbers of eyespots on the hindwings may be advantageous if they deflect predator attacks away from the forewings which are more important for flight.”

We have amended the text as suggested.

Edited Text	Original Text
-------------	---------------

Lines 66-68: ‘Second, higher numbers of eyespots on the hindwings may be advantageous if they deflect predator attacks away from the forewings which are more important for flight.’	‘Second, the lower forewing eyespot number could be maintained by predation pressure: butterflies with fewer forewing ventral eyespots may have a selective advantage.’
--	---

L85: Are spots likely to make the wing more conspicuous or do they dupe the predator into thinking they are striking at an important body part?

All the work done to date to differentiate between these two hypotheses have been conducted on large, intimidating eyespots (e.g. Stevens et al. 2008, 2009, De Bona et al. 2015) as opposed to small marginal eyespots such as those on our study species *Bicyclus anynana*. With small marginal eyespots, it remains largely unresolved whether they draw attacks to themselves by being more conspicuous *per se* or by mimicking an important body part. However, we have based our hypothesis on the conclusions of Prudic et al. (2015) who suggest that more conspicuous areas do attract more attacks by predators. We have now rephrased the text to make the basis from which we form our hypothesis clearer.

Edited Text	Original Text
Lines 84-86: ‘We hypothesise that, because the two additional forewing eyespots in Spotty Bicyclus anynana are likely to make this wing more conspicuous than the Wt forewing, more attacks would be directed at the forewing in Spotty [7], resulting in lower fitness.’	‘The two additional forewing eyespots in Spotty Bicyclus anynana are likely to make this wing more conspicuous than the Wt forewing. Hence, we hypothesise that more attacks would be directed at the forewing in Spotty, resulting in lower fitness.’

References

- De Bona, S., Valkonen, J. K., López-Sepulcre, A., & Mappes, J. (2015). Predator mimicry, not conspicuousness, explains the efficacy of butterfly eyespots. *Proceedings of the Royal Society B: Biological Sciences*, 282(1806), 20150202.
- Prudic, K. L., Stoehr, A. M., Wasik, B. R., & Monteiro, A. (2015). Eyespots deflect predator attack increasing fitness and promoting the evolution of phenotypic plasticity. *Proceedings of the Royal Society B: Biological Sciences*, 282(1798), 20141531.
- Stevens, M., Hardman, C. J., & Stubbins, C. L. (2008). Conspicuousness, not eye mimicry, makes “eyespots” effective antipredator signals. *Behavioral Ecology*, 19(3), 525-531.
- Stevens, M., Cantor, A., Graham, J., & Winney, I. S. (2009). The function of animal ‘eyespots’: conspicuousness but not eye mimicry is key. *Current Zoology*, 55(5), 319-326.

L94: delete “for”

This has now been deleted.

L133: not sure what you mean by “subject to similar numbers of male and female mantids being assigned to each form”

We are trying to convey here that each form, Wt and Spotty, were exposed to similar numbers of male and female mantids.

Edited Text	Original Text
Lines 132-134: 'For 20 cages (10 Wt and 10 Spotty), one mantid was introduced. Mantids were randomly assigned, whilst ensuring that each form was exposed to a similar number of male and female mantids.'	'For 20 cages (10 Wt and 10 Spotty), one mantid was introduced. Mantids were randomly assigned, subject to similar numbers of male and female mantids being assigned to each form.'

L140: delete "respectively"

This has now been deleted.

L250: not sure if "censored" is the word you are looking for

We have now replaced the word "excluded" with the word "censored".

L275: in control cages there was no difference in the wing damage...

We have now added "in wing damage" to the text.

Edited Text	Original Text
Lines 275-276: 'Second, in control cages, there was no difference in wing damage between Wt and Spotty, both on forewings (P = 0.80) and hindwings (P = 0.83).'	'Second, in control cages, there was no difference between Wt and Spotty, both on forewings (P = 0.80) and hindwings (P = 0.83).'

L281: Since most people use 0.05 as their cut-off for significance (arbitrary as it is) most people would consider this as being marginally insignificant not marginally significant. Perhaps try: "These differences border on significance for the forewings (P = 0.08...)"

We have now amended the text as suggested.

Edited Text	Original Text
Lines 282: 'These differences border on significance for the forewing...'	'These differences are marginally significant on the forewing...'

L300: again, this is not marginally significant, but bordering on significance.

This has now been rephrased here and in one other place in the ms.

Edited Text	Original Text
-------------	---------------

Lines 301-302: ‘The multinomial GLMM revealed a difference which borders on significance in the overall proportions of the four first strike categories between Wt and Spotty...’	‘The multinomial GLMM revealed a marginally significant difference in the overall proportions of the four first strike categories between Wt and Spotty...’
Lines 308-309: ‘...a difference bordering on significance between “forewing eyespots” (decreasing from 40% in Wt to 29.4% in Spotty) and “eyespots on both wings” ($P = 0.085$).’	‘...a marginally significant difference between “forewing eyespots” (decreasing from 40% in Wt to 29.4% in Spotty) and “eyespots on both wings” ($P = 0.085$).’

L302-308: This is all one sentence and is almost impossible to follow. Break it up.

We have now broken up the text indicated to improve its readability.

Edited Text	Original Text
Lines 303-309: ‘The pairwise comparisons showed two main conclusions (figure 3c). First, a significant difference in the relative proportion between “hindwing eyespots” (reducing steeply from 17.1% in Wt to 2.9% in Spotty) and “eyespots on both wings” (more than doubling from 17.1% in Wt to 38.2% in Spotty) ($P = 0.031$), suggesting that the mantids shifted from attacking the hindwings alone to attacking both wings simultaneously. Second, a difference bordering on significance between “forewing eyespots” (decreasing from 40% in Wt to 29.4% in Spotty) and “eyespots on both wings” ($P = 0.085$).’	‘The pairwise comparisons showed (figure 3c): (i) a significant difference in the relative proportion between “hindwing eyespots” (reducing steeply from 17.1% in Wt to 2.9% in Spotty) and “eyespots on both wings” (more than doubling from 17.1% in Wt to 38.2% in Spotty) ($P = 0.031$), suggesting that the mantids shifted from attacking the hindwings alone to attacking both wings simultaneously; and (ii) a marginally significant difference between “forewing eyespots” (decreasing from 40% in Wt to 29.4% in Spotty) and “eyespots on both wings” ($P = 0.085$).’